

# Laser ablation aerosol particle time-of-flight mass spectrometer (LAAPTOF): Performance, reference spectra and classification of atmospheric samples

Xiaoli Shen[1,2], Ramakrishna Ramisetty[1], Claudia Mohr[1,3], Wei Huang[1,2], Thomas Leisner[1], Harald Saathoff[1,*]

[1]Institute of Meteorology and Climate Research (IMK-AAF), Karlsruhe Institute of Technology (KIT), Hermann-von-Helmholtz-Platz 1, 76344 Eggenstein-Leopoldshafen, Germany
[2]Institute of Geography and Geoecology (IfGG), Karlsruhe Institute of Technology (KIT), Kaiserstr.12, 76131 Karlsruhe, Germany
[3]Now at: Department of Environmental Science and Analytical Chemistry, Stockholm University, Stockholm, 11418, Sweden

*Correspondence to*: Harald Saathoff (harald.saathoff@kit.edu)

**Abstract.** The laser ablation aerosol particles time-of-flight mass spectrometer (LAAPTOF, Aeromegt GmbH) is able to identify the chemical composition and mixing state of individual aerosol particles, and thus is a tool for elucidating their impacts on human health, visibility, ecosystem and climate. The overall detection efficiency (ODE) of the instrument we use was determined to range from ~(0.01 ± 0.01)% to ~(6.57 ± 2.38)% for polystyrene latex (PSL), ammonium nitrate ($NH_4NO_3$), and sodium chloride (NaCl) particles in the size rage of 200 to 2000 nm. Reference mass spectra of 32 different particle types relevant for atmospheric aerosol (e.g. pure compounds $NH_4NO_3$, $K_2SO_4$, NaCl, oxalic acid, pinic acid, and pinonic acid; internal mixtures of e.g. salts, secondary organic aerosol, and metallic core-organic shell particles; more complex particles such as soot and dust particles) were determined. Our results show that internally mixed aerosol particles can result in spectra with new clusters of ions, rather than simply a combination of the spectra from the single components. An exemplary one-day ambient data set was analysed by classical Fuzzy-clustering leading to six different particle classes. Correlating these particle classes with the reference spectra as well as direct comparison of the ambient data with the reference spectra has proven how useful they are for the interpretation of field measurements, for e.g. grouping data, and identifying special particle types and potential sources.

## 1 Introduction

Atmospheric aerosol particles impact visibility, interact with trace gases, can act as cloud condensation and ice nuclei, and influence the Earth's radiation budget (Seinfeld and Pandis, 2006). Especially the continuously evolving chemical composition of aerosol particles is of scientific interest, as it influences all aerosol effects (Burkholder et al., 2017; Pöschl, 2005). However, large knowledge gaps still exist related to the chemical composition of the organic and inorganic components and their mutual interaction (Jimenez et al., 2009; Murphy et al., 2006; Schill and Tolbert, 2013; Zhang et al., 2007).

Aerosol particles can contain various components ranging from volatile (e.g. nitrate, sulphate, ammonium salts, and many organic compounds), to refractory species (e.g. elemental carbon, minerals, and sea salt) (Pratt and Prather, 2012). The global aerosol mass burden was estimated to consist of 73.6% dust, 16.7% sea salt, 2.8% biogenic secondary organic aerosols (SOA), 2.3% primary organic aerosols (POA), 1.3% sulphate, 1.3% ammonium, 1.2% nitrate, 0.4% black carbon (soot), 0.2% anthropogenic SOA, and 0.2% methane sulphonic acid (Tsigaridis et al., 2006). SOA is estimated to account for the major



fraction of the total organic aerosol mass with dicarboxylic acids, such as oxalic acid suggested to be the main contributors
(Ervens et al., 2004). Ambient aerosols, either directly emitted (primary aerosols) or formed in the atmosphere (secondary
aerosols) from oxidation of gas-phase precursors or chemical reactions on particles, have typical lifetimes ranging from hours to
a few weeks (Pöschl, 2005). During their lifetime, the complexity of their chemical composition usually increases by coagulation,
cloud processing, or chemical reactions: Sea salt, POA, soot, or dust particles can e.g. heterogeneously react with secondary
organic compounds like organic acids and secondary inorganic compounds like sulfuric or nitric acid (Seinfeld and Pandis, 2006;
Usher et al., 2003). This modifies the particles' mixing state, with both internal (individual particles consisting of mixed
compounds, e.g. coating structures) and external mixtures (e.g. mixture of particles consisting of different compounds) (Li et al.,
2016). This underscores the importance of measuring aerosol chemical composition and its changes on short timescales and on a
single particle basis, which can be realized by on-line mass spectrometry.
One-line mass spectrometry includes bulk and single-particle measurements (Pratt and Prather, 2012). Single particle mass
spectrometry, which can be dated back to the 1970s, aims at in situ and real time identification of the chemical composition of
individual aerosol particles, hereby elucidating a particle's external and internal mixing properties (Noble and Prather, 2000).
Online single particle mass spectrometers (SPMS) commonly use pulsed lasers for particle desorption and ionization (LDI), with
the advantage of ionizing nearly all atmospheric particle components, including both non-refractory and refractory materials
(Kulkarni et al., 2011). To the best of our knowledge, so far no SPMS analysis is yet capable of providing a quantitative
composition analysis, since the ablation/ionization laser cannot interact with the entire particle, and the resulting ion fragments
and clusters are susceptible to matrix effects. In addition, ionization mechanisms are not fully understood (Murphy, 2007). The
first commercial SPMS combined LDI with a Time-of-Flight Mass Spectrometer (aerosol time-of-flight mass spectrometer,
ATOFMS, TSI GmbH) (Gard et al., 1997; Su et al., 2004). Several other home-build research SPMS were developed, each with
different advantages: Particle Analysis by Laser Mass Spectrometry (PALMS) (Murphy, 2007; Murphy and Thomson, 1995),
Laser Mass Analyser for Particles in the Airborne State (LAMPAS) (Trimborn et al., 2000), Single Particle Analysis and Sizing
System (SPASS) (Erdmann et al., 2005), Single Particle Laser Ablation Time-of-Flight Mass Spectrometer (SPLAT) (Zelenyuk
and Imre, 2005; Zelenyuk et al., 2009), Aircraft-based Laser ABlation Aerosol Mass spectrometer (ALABAMA) (Brands et al.,
2011), and Single Particle Laser Ablation Mass Spectrometer (SPLAM) (Gaie-Levrel et al., 2012) to name some of them. SPMS
have identified many different ambient particle types in different regions of the atmosphere, such as an elemental carbon/organic
carbon (ECOC), organic-sulphate, aged sea salt, biological, soil dust, and different metal dominated types (Dall'Osto et al., 2016;
Moffet et al., 2008; Murphy et al., 2006; Schmidt et al., 2017). These measurements all confirmed the complexity of individual
particles' mixing state, and demonstrated the usefulness of single particle mass spectra for apportionment of individual particle
sources, including e.g. fossil fuel and biomass burning combustion, cooking, marine, and shipping sources (Arndt et al., 2016;
Schmidt et al., 2017).
Currently, the only commercially available SPMS is the Laser Ablation Aerosol Particles Time-of-Flight mass spectrometer
(LAAPTOF, Aeromegt GmbH). It uses two laser diodes (wave length 405 nm, ~40 mW, ~50 μm beam spot diameter) for optical
counting and size recording by light scattering, and one excimer laser (ArF , 193 nm, ~ 4 mJ) for one step ablation/ionization.
The overall detection efficiency (ODE) of this instrument, defined as the number of single particle mass spectra obtained from
the total number of aerosol particles in the sampled air, was determined to range from ~0.15% to ~2.2% for polystyrene latex
(PSL) particles with geometric diameters ($d_p$) between 350 nm and 800 nm (Gemayel et al., 2016; Marsden et al., 2016). The
instrument used by Gemayel et al. (2016) exhibited a maximum ODE of ~2.2% for PSL particle diameters of 450 nm, while ~1%
at 600 nm was the peak ODE reported by Marsden et al. (2016), but only after modification of the instrument. The response of





the LAAPTOF to spherical PSL particles smaller than 350 nm and bigger than 800 nm, and the response to other particle types
with different shapes, have not been investigated systematically. The scattering efficiency (SE), defined as the number
percentage of particles detected by light scattering compared to the number of particles in the sampled air in front of the
aerodynamic inlet lens (ADL) of the instrument (cf. Fig. 1), is determined by the laser diodes, the detection optics, as well as the
photomultiplier tubes (PMT), and has a strong influence on the ODE of the instrument. Therefore, several groups tried to
improve this part of the instrument. Marsden et al. (2016) modified the detection stage geometry by replacing the detection laser
with a fiber coupled 532 nm, 1 W Nd:YAG solid state laser system with a collimated laser beam, accomplishing an order of
magnitude improvement in light detection sensitivity to PSL particles with 500–800 nm diameter. Zawadowicz et al. (2017)
modified the optical path of the laser diodes with a better laser beam of <1 mrad full angle divergence and 1000 μm detection
beam spot size, and applied light guides to enhance the scattered light collection, resulting in 2−3 orders of magnitude
improvement in optical counting efficiency to PSL particles with 500−2000 nm vacuum aerodynamic diameter ($d_{va}$). There are
only very few studies so far that discuss mass spectral patterns of different particle types measured by LAAPTOF. Gemayel et al.
(2016) presented spectra from ambient particles collected in the city centre of Marseille, France; pure soot and SOA coated soot
particles (positive spectra only; Ahern et al., 2016). Spectra from potassium rich feldspar, soot, Argentinian soil dust, and
Snomax (commercial ice nuclei) were shown by (Zawadowicz et al., 2017), and PSL and potassium rich feldspar spectra were
measured by Marsden et al. (2017). Reitz et al. (2016) presented peak assignments for pure ammonium nitrate and sulphate
particles, as well as for ambient particles measured at a suburban site of Düsseldorf, Germany, but did not show any spectra.
Marker ions generated from SPMS are likely instrument specific, as pointed out by Schmidt et al. (2017). Therefore, there is a
need for publicly available spectral information of this relatively new instrument.
There exists several techniques to group the large number of individual particle types and spectra resulting from SMPS
measurements, such as k-means, c-means and hierarchical clustering algorithms, neural network based methods such as ART2-A,
as well as the most recent algorithm of ordering points to identify the clustering structure (OPTICS), to help analyse the data
(Hinz et al., 1999; Murphy et al., 2003; Reitz et al., 2016; Zelenyuk et al., 2006b; Zhao et al., 2008). For LAAPTOF data
analysis, the Fuzzy c-means algorithm is commonly used to do classification based on the similarities of the individual spectra.
The number of the classes is chosen manually (Hinz et al., 1999; Reitz et al., 2016). There also exist target (reference
spectra/predefined clusters)-oriented methods that are used for analysing single particle mass spectrometer data, especially for
ambient monitoring (Hinz et al., 1999; Gleanta GmbH; LAAPTOF AnalysisPro, Aeromegt GmbH).
In this paper we have characterized our LAAPTOF instrument with respect to its ODE for PSL, $NH_4NO_3$, and sodium
chloride (NaCl) particles for a wide size range ($d_m$: 200−2000 nm PSL; 300−1000 nm $NH_4NO_3$ and NaCl). We present
laboratory based reference spectra for aerosol particles containing atmospherically relevant major components, which were
grouped in three categories: 1) particles consisting of pure compounds, e.g. $NH_4NO_3$, $K_2SO_4$, and organic acids; 2) particles
consisting of well-defined mixtures of pure salts and mixtures of organic compounds, e.g. α-pinene SOA, PSL internally mixed
with $K_2SO_4$, and other core-shell type of particles; and 3) particles consisting of complex mixtures, e.g. soot and dust particles.
These reference spectra may provide also other users comprehensive references for comparison purposes, and thus better
interpretation of ambient data. A one-day example of field data interpretation based on these reference mass spectra will be given
in chapter 3.3 and compared to a Fuzzy clustering approach.



## 2 Methods

### 2.1 LAAPTOF

The LAAPTOF has been described in several recent publications (Ahern et al., 2016; Gemayel et al., 2016; Marsden et al., 2016, 2017; Reitz et al., 2016; Zawadowicz et al., 2017). Therefore, we only briefly review the general operation steps that yield size and composition information of individual aerosol particles. The LAAPTOF instrument used in this study was delivered in April 2015 and may differ in a few technical aspects from earlier or later versions. A schematic of the main LAAPTOF components is given in Fig. 1 Particles with a vacuum aerodynamic diameter ($d_{va}$) between ~70 nm and 2.5 µm are sampled with a sampling flowrate of ~80 standard cubic centimetre per minute (SCCM), focused and accelerated by an aerodynamic lens, ADL (LPL-2.5, Aeromegt GmbH) with close to 100% transmission efficiency for particles with $d_{va}$ 100 nm to 2 µm, then pass through the particle time-of-flight (PTOF) chamber in which the individual particle can be detected by two sizing laser beams (405 nm continuous wave, 40 mW) separated by 11.3 cm. Based on the particle time of flight between the two laser beams, its $d_{va}$ can be determined and recorded. After detection by the second sizing laser, a nanosecond (ns) excimer laser pulse (wave length: 193 nm, pulse duration: 4 to 8 ns, maximum pulse energy: ~8 mJ, beam diameter: ~ 300 µm when it hits the particle, power density: ~$10^9$ W·cm$^{-2}$, ATLEX-S, ATL Lasertechnik GmbH) can be triggered to desorb and ionize particle compounds. A laser pulse energy of 4 mJ was used for all the measurements in this study. More details about the ionization region geometry are given by Ramisetty et al. (2017). The resulting ions are analysed by a bipolar time-of-flight mass spectrometer (BTOF-MS; TOFWERK AG; mass resolution of m/ △ m ~600 to 800 at 184 Th, mass range m/q=1 up to ~2000 Th). The resulting cations and anions are detected by corresponding microchannel plate arrays (MCPs), producing a pair of positive and negative spectra for each single particle.

For each type of laboratory generated aerosol particle, we measured at least 300 mass spectra. Data analysis is done via the LAAPTOF Data Analysis Igor software (Version 1.0.2, Aeromegt GmbH). There are five main steps for the basic anaylsis procedure: a) removal of the excimer laser ringing signal from the raw mass spectra; b) determination of the signal baseline; c) filtering for empty spectra; d) mass calibration; and e) stick integration. Spectra-to-spectra differences in peak positions due to variance in the position of particle-laser interaction complicate the mass calibrations. Details can be found in the supplementary information (SI). Spectra presented in this paper were typically normalized to the sum of ion signal before further aggregation.

For the grouping of ambient data, we used two different classification methods. The Fuzzy c–means clustering algorithm is embedded in the LAAPTOF Data Analysis Igor software and starts from random class centres. Particle spectra with a minimum distance between their data vectors and a cluster centre will be grouped into this specific class (Hinz et al., 1999). Since each spectrum can belong to multiple classes (Reitz et al., 2016) the resulting fraction/percentage for each class represents the information about the degree of similarity between aerosol particles in one particular class, and not a number percentage. The second method developed in this study is based on the correlation between each ambient spectrum and our reference spectra. The resulting Pearson's correlation coefficient (r) is used as the criteria to group particles into different types (here we use "types" instead of "classes" in order to differentiate these two classification methods). When r is above the threshold value 0.6, the ambient spectrum is considered to have high correlation with the corresponding reference spectra. For simplification we chose 10 positive and 7 negative reference spectra. For example, we only use German soil dust as the reference for arable soil dust rather than using four arable soil dust samples from different places. More details about the procedure for this method as well as the corresponding equations and uncertainties estimation can be found in the supplementary information.

### 2.2 Aerosol particle generation





The aerosol particles measured in this study (Table S1) were generated in three different ways (cf. Fig.1). Samples for pure
particles (except SiO₂) and homogeneous and heterogeneous mixtures (except SOA) were dissolved in purified water and
nebulized (ATM 221; Topas GmbH) with dry synthetic air, passed through two diffusion dryers (cylinder filled with Silica gel,
Topas GmbH), and then size selected by a Differential Mobility Analyser (DMA 3080, TSI GmbH) before being sampled by
LAAPTOF (setup A). A condensation particles counter (CPC 3010, TSI GmbH) was used to record the particle number
concentration. SOA particles from ozonolysis (~6 ppm ozone) of α-pinene (~2.2 ppm), a common laboratory-based surrogate for
biogenic SOA (Saathoff et al., 2009), were formed in the 3.7 m³ stainless steel Aerosol Preparation and Characterization (APC)
chamber and then transferred into the 84.5 m³ simulation chamber AIDA (Aerosol Interactions and Dynamics in the Atmosphere)
of KIT (Saathoff et al., 2003). Soil dust samples were dispersed by a rotating brush generator (RGB1000, PALAS) and injected
via cyclones into the AIDA chamber. Sea salt particles were generated in different ways (Wagner et al., 2017) and sampled from
the AIDA chamber (setup B in Fig. 1). Soot particles from incomplete combustion of propane were generated with a propane
burner (RSG miniCAST; Jing Ltd.) and injected into and sampled from a stainless steel cylinder of 0.2 m³ volume. Ambient
aerosol particles from a rural site near Leopoldshafen, Germany (refer to section 2.3) were sampled through a PM₂.₅ inlet (SH 2.5
- 16, Comde-Derenda GmbH) with 1 m³ h⁻¹, a fraction of which was guided into the LAAPTOF (set up C in Fig. 1). Silica,
Hematite, Illite_NX, mineral dust, black carbon from Chestnut wood (University of Zürich, Switzerland), and urban dust and
diesel soot reference particles from NIST, were directly sampled from the headspace of their reservoirs.
**2.3 Field measurement**
Unusually high particle number concentrations, similar to downtown Karlsruhe (a city in southwest Germany), were observed
frequently northeast of Karlsruhe by particle counters on-board a tram wagon (www.aero-tram.kit.edu) intersecting the city
(Hagemann et al., 2014). To study the nature and to identify possible sources of these particles, their number, size, chemical
composition, associated trace gases and meteorological conditions were measured from July 15th to September 1st, 2016 at a rural
location (49°6'10.54"N, 8°24'26.07"E), next to the tram line north of the village of Leopoldshafen, Germany. LAAPTOF
measurements provided information on size and mass spectral patterns for individual particles. In this paper we use data from
one day as an example for the potential interpretation of LAAPTOF spectral data using reference spectra.
**3 Results and Discussion**
**3.1 Determination of LAAPTOF performance parameters**
**3.1.1 Scattering efficiency, hit rate and overall detection efficiency for standard samples**
In the literature there are two definitions of detection efficiency (DE) of SPMS used: one is equal to the scattering efficiency (SE)
of the detection lasers (Brands et al., 2011; Gaie-Levrel et al., 2012; Su et al., 2004; Zelenyuk and Imre, 2005; Zelenyuk et al.,
2009), while the other one is the product of SE and hit rate (HR) of the ablation/ionization laser (Su et al., 2004; Gemayel et al.,
2016; Marsden et al., 2016). The hit rate (HR) is the fraction of particles detected by the scattering optics leading actually to a
useful mass spectrum. In this paper we use overall detection efficiency (ODE), defined by the following equations:

$$ODE = SE \times HR \times 100\% \tag{1}$$

$$SE = N_d/N_0 \times 100\% \text{ (transmission efficiency of ADL is included)} \tag{2}$$



$$HR=N_s/N_d \times 100\% \text{ (ionization efficiency is included)} \qquad (3)$$

$$N_0= C_n \times \text{flowrate} \times \text{time} \qquad (4)$$

where $N_d$ is the number of particles detected by light scattering, $N_0$ is the number of particles in front of the ADL, $N_s$ the number of bipolar spectra, and $C_n$ is the particle number concentration (cm$^{-3}$) measured by a CPC in front of the ADL. The sample flowrate of the LAAPTOF is ~80 cm$^3$ min$^{-1}$.

HR, SE, and ODE for spherical PSL particles as a function of electrical mobility equivalent diameter $d_m$, are plotted in Fig. 2. It should be noted that the LAAPTOF detection behaviour may vary depending on the alignment of the ADL and the optical components (especially the detection laser diodes), which is difficult to reproduce. We therefore show results for PSL particles based on 2 repeated experiments after 3 alignments each, and thus a total of 6 experiments for each data point. The uncertainty intervals in Fig. 2 are the difference between the maximum/minimum and the average values obtained from these 6 experiments. As shown in panel A of Fig. 2, for particle diameters from 200 to 400 nm, HR$_{PSL}$ exhibits an increase from 69% to 94%, decreases to 83% for 700 nm particles, and then becomes stable at ~85% for particles with diameters up to 2 μm. The average HR$_{PSL}$ ($\overline{HR}_{PSL}$) is ~84%. SE$_{PSL}$ and ODE$_{PSL}$ show an M-like shape with two peaks, at 500 nm (SE$_{PSL}$ 3.0%, ODE$_{PSL}$ 2.7%), and at 1000 nm (SE$_{PSL}$ 4.8%, ODE$_{PSL}$ 4.2) (see panel B and C of Fig. 2). We attribute this behaviour to a combined effect of the spherical shape of PSL particles and the optical system of this instrument, e.g. Mie resonances related to particle size and laser wavelength (see section 3.1.2 for details). As shown in panel C of Fig. 2, values and trends of ODE$_{PSL}$ in the size range of 300−800 nm of our instrument are similar to those reported by (Gemayel et al., 2016)and (Marsden et al., 2016) for their LAAPTOF instruments. A recent LAAPTOF study by Zawadowicz et al. (2017) shows comparable results for PSL particles with $d_p \leq 500$ nm, and an M-like shape of ODE in the size range of 200−2000 nm (after instrument modification).

We also measured mass spectra of non-spherical NH$_4$NO$_3$ (χ=0.8, Williams et al., 2013) and NaCl particles (cubic, χ=1.06 to 1.17, Zelenyuk et al., 2006a). Similar as for PSL particles, NH$_4$NO$_3$, and NaCl particles show relatively high and stable HR with average values of 80% and 66% (see panel D in Fig. 2), thus SE and ODE have a similar trend. No M-like shape of ODE as a function of particle size is observed due to the different light scattering properties of the non-spherical salt particles (Bohren and Huffman, 2007) (see panels E and F in Fig. 2). Comparable results were shown for (NH$_4$)$_2$SO$_4$ particles (χ=1.03 to 1.07, Zelenyuk et al., 2006a) by Zawadowicz et al. (2017). As shown in Fig. 2 E–F, SE and ODE decrease with increasing shape factor for salt particles of the same size. We will discuss this in more detail in the following section.

### 3.1.2 Factors influencing overall detection efficiency

There are various factors that can influence the ODE. One of these is particle size. For particles with diameters below 200 nm, the scattered light becomes too weak to be detected due to the strong dependence of the scattering intensity on particle size (Bohren and Huffman, 2007). For particles with diameters larger than 2 μm, focusing by the ADL is much less efficient, resulting in a higher divergence of the particle beam. This lowers the probability of larger particles to be detected by the detection/scattering laser and/or to be hit by the ionization laser. In addition, light scattering of spherical particles like PSL changes from Rayleigh to Mie to Geometric scattering as the size parameter $\alpha=\pi d_p/\lambda$ increases from <<1 to ~1 to >>1 (Seinfeld and Pandis, 2006). α ranges from ~1.5 to 19 for 200−2500 nm PSL particles, and is thus in the Mie scattering regime and the reason for the M-like shape of SE$_{PSL}$ and ODE$_{PSL}$. As long as the particle diameter ($d_p$) is smaller than the wavelength of the detection laser light, here 405 nm, the scattered radiation intensity (proportional to $d_p^6$) will rapidly decrease with decreasing particle sizes, resulting in low ODE. ODE is e.g. 0.01% for 200 nm PSL particles. For non-spherical particles like salts, their SE



and ODE are also size dependent (panel F in Fig. 2), due to size-dependent light scattering ability and particle beam divergence.
However, they don't exhibit Mie resonance, and thus don't show an M-like shape in their scattering efficiency.
Optical properties of the particles have a strong impact on how light is scattered and absorbed, and thus also greatly influence
scattering efficiency and ionization efficiency (or hit rate), respectively. As shown in Fig. 2, ODE for $NH_4NO_3$ is higher than that
for NaCl at any size we studied (panel F). This is caused by differences in their optical properties and shapes. The reference
spectra of pure $NH_4NO_3$ and $(NH_4)_2SO_4$ particles showed intensive prominent peaks for pure $NH_4NO_3$ particles but only one
weak peak for pure $(NH_4)_2SO_4$ particles. This is indicating that $NH_4NO_3$ is a better absorber than $(NH_4)_2SO_4$, and thus easier to
ablate and ionize. For homogeneous mixtures of these two ammonium salts, the sulphate species are detected much more easily
due to increased light absorption by the nitrate component (refer to section 3.2.2). Soot particles are good light absorbers and
thus relatively easy to ablate and ionize. However they scatter only little light due to the small size (typically ~ 20 nm) of the
primary particles forming their agglomerates, and are thus hardly detected. Their usually small size is an additional disadvantage
for their detection. Some small organic compounds with weak absorption properties are hard to ablate and ionize as well, e.g.
oxalic acid ($C_2H_2O_4$), pinic and cis-pinonic acids measured in this study had much weaker signals in the spectra (~80% lower)
than macromolecular organic compounds in PSL or humic acid particles.
Particle morphology is another important factor. The scattering efficiency for non-spherical $NH_4NO_3$ is higher than for
spherical PSL particles in the size rage of 300−800 nm (Fig. 2 B−E) (Ackerman et al., 2015). For larger particle sizes ($d_m$> 800
nm), beam divergence offsets the shape effect (Murphy, 2007). Apart from that, the increase of surface roughness and
inhomogeneity can promote the scattering capability of particles (Ackerman et al., 2015).
The incident intensity of radiation, which is another parameter that influences the light scattered by particles (as well as
background signal caused by stray light), is related to power and beam dimensions of the laser. A laser power of 40 mW was
used in this study. Marsden et al. (2016) replaced the detection laser with a fibre coupled 532 nm, 1 W Nd:YAG solid state laser
system that has a collimated laser beam, resulting in an order of magnitude improved sensitivity to PSL particles with 500−800
nm diameter. Zawadowicz et al. (2017) used laser diodes with a laser beam of <1 mrad full angle divergence and 1000 μm
detection beam spot size, and applied light guides to enhance the scattered light collection, resulting in 2−3 orders of magnitude
improvement in optical counting efficiency of PSL particles with $d_{va}$ 500−2000 nm. In addition, alignment of the excimer laser
focus in x, y, and z position influences optimum hit rates (Ramisetty et al., 2017).
There are further instrumental aspects that affect the detection efficiency. High number concentrations of the incoming
particles influence the ODE, since there can be more than one particle present between the two detection lasers. The transmission
efficiency of the ADL is included in the scattering efficiency, and thus directly influences it. The size range of particles focused
in the lens, and the particle beam width strongly depend on the configuration of the ADL (Canagaratna et al., 2007; Johnston,
2000). Liu lenses and Schreiner lenses can focus the particles in the size range of 80−800 nm, and 300−3000 nm, respectively
(Kamphus et al., 2008; Liu et al., 1995; Schreiner et al., 1999). The ADL transmission efficiency of our instrument, as
determined by the manufacturer (Aeromegt GmbH), is close to 100% for particles with $d_{va}$ 100–2000 nm.
**3.2 LAAPTOF reference spectra of laboratory generated particle types**
Particles for which reference spectra are presented here are listed in Table S1. For each type of these aerosol particles, we present
averaged spectra for typically 300 to 500 single particles. The relative standard deviations (RSD, SD normalized to signal) for
the characteristic peaks are in the range of 15−186%, median value 77%.



Despite the lack of full quantitativeness of the LAAPTOF, mass spectral signal amplitudes show an increase with particle
size. However, no systematic changes in the mass spectral signatures were observed for different particle sizes. Therefore,
particles in the optimum size range of the LAAPTOF ($d_m$ = 800 nm) and with good signal-to-noise ratio were chosen to generate
reference spectra. For polydisperse particles generated in the AIDA chamber, the corresponding average spectra include particles
of broader size distributions compared to those preselected by the DMA. Information on particle generation or source as well as
the sizes is listed in Table S1.
A qualitative comparison between the relative peak intensity ratios within an single particle spectrum and those in another
spectrum can yield relative quantitation information, as suggested by Gross et al. (2000). We add information on typical peak
ratios to some of our reference spectra to help identify specific species.
**3.2.1 Pure compound particles**
Although particles consisting of one single species only are rarely sampled in the atmosphere, interpretation of mass spectra of
ambient samples is supported by the knowledge about the mass spectra of pure compounds. In the following mass spectra for a
few typical ambient aerosol constituents are discussed.
Figure 3 shows average spectra for pure compound aerosol particles. For $NH_4NO_3$ particles (panel A), we observed the
positive ions m/z 18 $NH_4^+$ and m/z 30 $NO^+$; and the negative ions m/z 46 $NO_2^-$ and m/z 62 $NO_3^-$, similar to Reitz et al. (2016).
The LAAPTOF is much less sensitive to ammonium than nitrate fragments, leading to a weak $NH_4^+$ signal and prominent $NO^+$,
$NO_2^-$ and $NO_3^-$ peaks. The ratio of $NO^+$ to $NH_4^+$ is ~48, and the ratio of $NO_2^-$ to $NO_3^-$ is ~4. The prominent peak of $NO^+$ arises not
only from nitrate (majority), but also from ammonium (Murphy et al., 2006). In our ammonium nitrate spectra, there are weaker
signatures of m/z 46 $NO_2^+$ and m/z 125 $HNO_3 \cdot NO_3^-$ (not shown here, but visible and reproducible), which were also observed in
PALMS mass spectra (Zawadowicz et al., 2015). For $K_2SO_4$ particles, we observed the potassium signals at m/z 39 $K^+$ and m/z
41 $K^+$, and a sulphate signature with ion clusters grouped around m/z 32 $S^-$, m/z 64 $SO_2^-$, m/z 80 $SO_3^-$ and m/z 96 $SO_4^-$. Note that
peaks with high intensity exhibit "ringing" in the raw spectra, resulting in small peaks beside the main ones in the integrated
stick spectra, such as m/z 40$^+$ besides m/z 39 $K^+$ in Fig. 3 (B), and m/z 36$^-$ besides m/z 35 $Cl^-$ in the spectra for sodium chloride
NaCl (Fig. S1). Therefore, the real intensities of m/z 39 $K^+$ and m/z 35 $Cl^-$ should include their corresponding side ringing peaks.
The ratio of m/z 39 $K^+$ to m/z 41 $K^+$ is ~13.2, close to the natural isotopic ratio of ~13.9 for $^{39}K/^{41}K$. For pure NaCl particles, the
ratio of m/z 35 $Cl^-$ to m/z 37 $Cl^-$ is ~3.2, similar to the natural isotopic ratio of ~3.1 for $^{35}Cl/^{37}Cl$. Therefore, these two isotopic
ratios can be used as markers to identify K and Cl measured by LAAPTOF. Another inorganic compound measured here is silica
(Fig. S2) with the typical peak ratio (~1.0) of m/z 76 $SiO_3^-$ + m/z 77 $HSiO_3^-$ to m/z 60 $SiO_2^-$. The corresponding histograms of
such ratios for different particle samples can be found in Fig. S3.
High signal intensities in oxalic acid spectra are observed at m/z 18 $H_2O^+$, 28 $CO^+$, and 30 $CH_2O^+$, as well as some weaker
peaks at m/z 40$^+$, 44$^+$, 56$^+$, and 57$^+$. M/z 89 $C_2O_4H^-$ is used as signature ion for oxalic acid in other SPMS studies (Roth et al.,
2016). In our study, a distinct signal at around m/z 89$^-$ is observed as well, indicating oxalate fragment formation after laser
ablation.
In order to identify humic like substances in the ambient particles, we measured humic acid particles (Fig. S4) and found
hydrocarbon and elemental carbon fragments, with very prominent peaks at m/z 24$^-$, 25$^-$, and 26$^-$ suggested to be organic ions
(Silva et al., 2000), as well as peaks at m/z 25$^-$, 26$^-$, 49$^-$, and 73$^-$ for unsaturated organic compounds.
**3.2.2 Particles consisting of well-defined internal mixtures**





Figure 4 shows average spectra from homogeneously internally mixed particles. The spectrum from the mixture of $NH_4NO_3$ and
$(NH_4)_2SO_4$ (panel A) contains the signature from pure $NH_4NO_3$ particles, but with lower relative intensities (each peak intensity
is normalized to the sum of ion signal) for $NO_2^-$ and $NO_3^-$, due to the formation of anion clusters at ~m/z=80 $SO_3^-$ and 97 $HSO_4^-$.
In addition, compared to the pure $NH_4NO_3$ particles, the ratio of $NO^+$ to $NH_4^+$ (~34) is ~30% lower in the spectrum for the
mixture, due to its lower molar ratio of nitrate/ammonium, whereas the ratio of $NO_2^-$ to $NO_3^-$ (~7) is 80% higher. Nitrate is
believed to assist in light absorbing for the mixed particles, resulting in a sulphate signature that could not be observed for pure
$(NH_4)_2SO_4$. This exemplifies potential effects of individual particle chemical composition on mass spectral performance of the
LAAPTOF. For the mixture of $K_2SO_4$ and NaCl (panel B), similar signatures as for the pure particles were observed. Compared
to the pure NaCl particle spectra, the signal intensity of $Na^+$ is decreased. This can be explained by more cations formed from the
mixed particles, including from potassium, which has a higher ionization potential and lower lattice energy than NaCl. For the
mixed particles, expected clusters such as 113/115 $K_2Cl^+$, 109 $KCl_2^-$, and 119 $NaSO_4^-$ and a minor fragment 97 $KNaCl^+$ were
observed, but not 81/83 $Na_2Cl^+$ as found in pure NaCl particles. These results show that compared to pure compounds, mass
spectra from aerosol particles consisting of mixtures can feature new ions, while some marker ions for the pure compounds may
disappear. These spectra are thus not simply a combination of the spectra from single component particles. Another example for
an inorganic mixture of $NH_4NO_3$ and $K_2SO_4$ is provided in Fig. S5. The α-pinene SOA spectrum is shown in panel (C) of Fig. 4.
Ablation of α-pinene SOA particles forms different types of organic fragments: 1) hydrocarbon and oxygenated organic
fragments $C_xH_yO_z$, (x=1–6, y=0–9, z=0–3, details about the peak assignments can be find in Table S2), except for m/z 59$^+$, 83$^+$,
85$^+$, and 95$^+$, are comparable to the combination mass spectral patterns for cis-pinonic and pinic acids (refer to Fig. S6) which are
oxidation products from α-pinene ozonolysis (Saathoff et al., 2009; Yu et al., 1999); 2) Carbon clusters 12 $C^+$, 24 $C_2^+$, 36 $C_3^+$,
and 60 $C_5^+$, with the most prominent peak in 12$C^+$, assigned to both soot and organic matter fragments in another LAAPTOF
study (Ahern et al., 2016); 3) Carboxylic acid groups in the negative spectra, e.g. 45 $COOH^-$, 59 $CH_2COOH^-$, 73 $C_2H_4COOH^-$, 85
$C_3H_4COOH^-$ and 99 $C_4H_6COOH^-$.
Figure 5 (A) shows the spectrum for heterogeneously internally mixed $K_2SO_4$ and PSL particles (PSL core, $K_2SO_4$ shell). All
signatures for PSL particles, i.e. hydrocarbon fragments in positive spectra, intensive organic signature m/z 24$^-$, 25$^-$, and 26$^-$,
carbon clusters $C_n^{+/-}$, and m/z 49$^-$ and 73$^-$ fragments arising from unsaturated structures such as aromatic structures are retained in
this spectra (grey labels), and the corresponding peak intensities are similar to the pure PSL particles (refer to Fig. S7). However,
the intensities of most of the $K_2SO_4$ fragments weaker compared to pure $K_2SO_4$ particles, likely due to the quite thin or only
partial coating layer of $K_2SO_4$ on the PSL core (the nominal geometric size of the PSL particles mixed with the aqueous solution
of $K_2SO_4$ was 800 nm which is the same size that was selected by the DMA prior to sampling by the mass spectrometer.). The
most prominent peak at m/z 39$^+$ with a normalized intensity of ~0.46, containing both $K^+$ and $C_3H_3^+$ fragments, is mainly
attributed to $K^+$ (intensity ~0.73 for pure $K_2SO_4$), since the intensity of $C_3H_3^+$ (~0.06) for pure PSL is much lower (refer to Fig.
S9). The still intensive signal from 39 $K^+$ despite the weaker sulphate peaks corresponds to the high sensitivity of the instrument
for potassium. Fig. 5 (B) shows the average spectrum for poly(allylamine hydrochloride) coated gold particles. Prominent
signatures of nitrogen containing compounds (NOCs) is observed at m/z 58 $C_2H_5-NH-CH_2^+$, 15 $NH^+$, 26 $CN^-$, and 42 $CNO^-$, as
well as the signatures for unsaturated organic compounds at m/z 25$^-$, 26$^-$, 49$^-$, and 73$^-$. Strong intensities for m/z (35$^-$ plus 36$^-$)
and 37$^-$ with ratio a of ~3.1 can be assigned to Cl isotopes derived from the hydrochloride. We also observed small gold peaks at
m/z 197$^{+/-}$ both in positive and negative spectra.
Mass spectra for other well-defined compounds,, i.e. synthetic hematite and pure sea salt particles, are also provided in the
supplementary information (Fig. S8 and S9).



### 3.2.3 Particles consisting of complex mixtures

Figure 6 shows the average spectra for different types of soot particles. All of them show characteristic patterns for elemental
carbon (EC) $C_n^{+/-}$. For soot1 with high organic carbon (OC) content from propane combustion in the laboratory (panel B),
prominent peaks were observed at m/z 28 $CO^+$ and 27 $C_2H_3^+$, as well as some other organic carbon signatures at m/z $39^+$, $40^+$, $44^+$
and $56^+$. All the organic signatures in soot1 with high OC were also observed for soot3, lignocellulosic char from Chestnut wood
(panel D), indicating that biomass burning soot contains a significant fraction of OC. It should be noted that biomass burning will
also form potassium, thus m/z $39^+$ contains both $K^+$ and $C_3H_3^+$ fragments. M/z $24^-$, $25^-$ and $26^-$ can be observed in all the soot
types, but with a bit different patterns: 1) soot with high EC content shows very high m/z $24^-$ (~2 to 3 times than m/z $25^-$), while
2) soot with high OC shows comparable or even higher m/z $25^-$ to/than m/z $24^-$. These patterns might provide help to distinguish
EC and OC contributions in the spectra from ambient particles.
Figure 7 shows spectra for Arizona test dust (milled desert dust) (penal A), arable soil SDGe01 sampled from Gottesgabe in
Germany (B), and agricultural soil dust collected from harvesting machines after rye and wheat harvest (C). For Arizona test dust,
we observed high mineral signatures of aluminium and silicon containing clusters, namely 27 $Al^+$, 28 $Si^+$, 44 $SiO^+$, 43 $AlO^-$, 59
$AlO_2^-$, 60 $SiO_2^-$, 76 $SiO_3^-$, 119 $AlSiO_4^-$, 179 $AlSiO_4.SiO_2^-$, 136 $(SiO_2)_2O^-$. It should be noted that high 16 $O^-$ and 17 $OH^-$
accompany the intensive mineral signatures, attributed to the adsorbed water on the active surface of mineral particles. In
addition, other mineral related metal clusters, e.g. 7 $Li^+$, 23 $Na^+$, 24 $Mg^+$, 40 $Ca^+$, 39/41 $K^+$, 55 $Mn^+$, 56 $Fe^+$, 58 $Ni^+$, 64 $Cu+$,
metal oxides and hydroxides, 56 $CaO^+$, 57 $CaOH^+$, 96 $Ca_2O^+$, 112 $(CaO)_2^+$, and 88 $FeO_2^-$, as well as weak anion clusters of
organic signature (m/z 24 $C_2^-$, 25 $C_2H^-$, 26 $C_2H_2^-$, and 42 $C_2H_2O^-$), NOCs (m/z 26 $CN^-$ and 42 $CNO^-$), chloride (m/z 35 $^-$ and 37 $^-$),
sulphate (m/z $32^-$, $48^-$, $64^-$, $80^-$, and $97^-$), phosphate (63 $PO_2^-$ and 79 $PO_3^-$), diacids (oxalate 89 $(CO)_2OOH^-$ and 117$(CO)_3OOH^-$)
and an unknown fragment m/z 148- were observed in the spectra (A). M/z $26^+$ in panels (B) and (C) is much higher than m/z $24^-$
and $25^-$, due to the contribution of CN fragments from NOCs. Similar signatures can also be observed in the spectra for Saharan
desert dust (Fig. S10).
For soil dust, most of their mineral and organic fragments are similar as desert dust, however with different intensities, e.g.
m/z $24^-$, $25^-$, $26^-$, and $42^-$ (labelled in green) are more intensive than those in desert dust, indicating higher organic compound
content. Some peak ratios of fragments are similar across the different dust types, e.g. 40 $Ca^+$ to 56 $CaO^+$ is 2.2, 1.1, and 2 for
desert dust, arable soil dust and agricultural soil dust, respectively. Compared with desert dust, there are different fragments from
soil dust particles, e.g. EC patterns (labelled in grey), organic acids signatures (blue), ammonium signatures (orange),
unsaturated organic fragments (m/z $49^-$ and $73^-$) and some other unknown fragments (red). For arable soil dust particles, we also
measured samples from Paulinenaue in Germany (SDPA01), Argentina (SDAr08) and Wyoming in USA (SDWY01) (refer to
Fig. S11). Dominant mass spectral peak patterns are similar across all soil dust samples. They are located at around m/z $27^+$, $39^+$,
and $56^+$ in the positive spectra; and $26^-$, $42^-$, $60^-$, and $76^-$ in negative spectra. Less prominent but reproducibly detected are
carboxylic acid groups (e.g. $COOH^-$) and EC patterns. German soil dust, however, contains more organic species than the soil
dusts from Argentina and the USA according to the intensities of m/z $24^-$, $25^-$, and $26^-$, while Argentinean soil dust contains
much less mineral species comparing the mineral signatures e.g. $27^+$, $28^+$, $40^+$, $44^+$, and $56^+$. The ratios of m/z 39 $K^+$ and 41 $K^+$
(3.6, 3.8, 3.5, 5.3 for SDGe01, SDPA01, SDAr08, and SDWY01, respectively) are much lower than the typical peak ratio (~10.6)
for potassium (Table 1), indicating that they are likely contributed to by both potassium isotopes and hydrocarbon fragments.
For agricultural soil dust particles, obviously ammonium (m/z 18 $NH_4^+$ and 30 $NO^+$), phosphate (m/z 63 $PO_2^-$, 79 $PO_3^-$, and
95 $PO_5^-$) and potassium signatures (m/z 39 $K^+$ and 41 $K^+$) can be found in the spectra, attributed to fertilization. Apart from that,
typical biological signatures were observed: 1) the strong m/z $26^-$, $42^-$, and $39^+$ pattern is similar to the potassium organo-



nitrogen particle type observed by an ATOFMS at an urban site in Barcelona (Dall'Osto et al., 2016), and which were assigned to
carbohydrates, arising from biogenic species (Schmidt et al., 2017; Silva et al., 2000). 2) $26^-$ and $42^-$ could also be contributed by
CN$^-$ and CNO$^-$ derived from NOCs, i.e. amines, as well as m/z 30 CH$_3$NH$^+$, 58 C$_2$H$_5$NHCH$_2^+$, and 59 (CH$_3$)$_3$N$^+$. These biological
signatures have also been observed by ALABAMA in the field (Schmidt et al., 2017). 3) Some weak but reproducibly detected
fragment pattern at around m/z 77 C$_6$H$_5^+$, 91 C$_7$H$_7^+$, 103 C$_8$H$_7^+$, 105 C$_8$H$_9^+$, and 115 C$_9$H$_7^+$ might be originate from aromatic
compounds. Similar patterns can also be found for PSL particles (Fig. S7).
Other examples for complex mixtures, i.e. illite and sea salt particles with biological components are provided in the
supplementary information (Fig. S12 and S9).
All the peak assignments and mass spectral patterns like signature peaks as well as some stable peak ratios mentioned above
have been summarized in Table S2 and Table 1, respectively. We consider these laboratory-based reference spectra as useful for
the analysis of data obtained also by other LAAPTOF versions and to some extend even for other single particle mass
spectrometers. Similar mass spectra are to be expected as long as they use similar ablation & ionization laser pulses (4 mJ, 193
nm), inlet regions for the mass spectrometer, and mass spectrometer types. In the near future, we plan to make these laboratory-
based reference spectra publicly available via the EUROCHAMP-2020 data base (www.eurochamp.org).
**3.3 Interpretation of field data**
Figure 8 shows an example of bipolar mass spectra for six different particle classes measured in the field campaign at a rural site
near Leopoldshafen in southwest Germany. On July 29$^{th}$, 2016 within 24 hours, 7314 particles were detected, successfully
ablated and mass spectra generated by LAAPTOF. The 7314 pairs of spectra were then clustered by the Fuzzy c-means
algorithm, resulting in six classes. The resulting number of classes with clearly different features depends on the experience of
the operating scientist to identify them (please refer to the details of Fuzzy clustering procedure in the supplementary
information). The Fuzzy results are compared with the laboratory-based reference spectra by calculating their correlation
coefficients (cf. Fig. 9). All classes exhibit a sulphate signature with m/z 97 HSO$_4^-$ and m/z 80 SO$_3^-$; a nitrate signature with m/z
46 NO$_2^-$ and 62 NO$_3^-$; an organic compound signature with m/z 24 C$_2^-$, 25 C$_2$H$^-$, and 26 C$_2$H$_2$/CN$^-$; and a NOC signature with
m/z 26 CN$^-$ and 42 CNO$^-$ in the negative spectra. More characteristic signatures for each particle class can be observed in the
positive spectra. All particles measured on this day show a 35% similarity to class 5 with obvious signatures for potassium (K)
and sulphate, with significant correlation with the reference particles containing potassium and sulphate (Fig. 9). Besides, class 5
also has significant correlation with some other cations arising from ammonium, organic compounds, and dust. The ratio of m/z
$(39^+ + 40^+)$ to $41^+$ is ~11, close to the value for pure K$_2$SO$_4$ particles (~13.5), thus we assigned them to K$^+$ rather than organic
fragments. Further, there is a 15% similarity to class 4 with prominent ammonium signatures at m/z 18 NH$_4^+$ and 30 NO$^+$,
sulphate signatures, as well as a relatively weaker but reproducible nitrate signature. The corresponding spectrum is similar as
the spectrum for the homogeneous mixtures of NH$_4$NO$_3$ and (NH$_4$)$_2$SO$_4$ (panel A in Fig. 4). This class also has strong correlation
with both positive and negative reference spectra for the mixture of ammonium nitrate and ammonium sulphate particles.
Ammonium, nitrate and sulphate are the major secondary inorganic species in atmospheric aerosol particles (Seinfeld and Pandis,
2006), thus we name this class "secondary inorganic". It should be noted that this class has significant correlation with
ammonium and cations arising from oxalic acid, however class 4 has weak correlation with the signature cation, i.e. m/z 89
C$_2$O$_4$H$^-$ (oxalate), of oxalic acid. Therefore, we can rule out a significant contribution of oxalic acid. There is also a 15%
similarity to class 2 (sodium rich), with a characteristic pattern of a strong signal at m/z 23 Na$^+$ accompanied by two weaker
peaks at m/z 39 K$^+$ (with typical potassium peak ratio of ~12) and 63$^+$ (might contain both Cu$^+$ and C$_5$H$_3^+$ fragments). Class 2 has





significant correlation with the cations (i.e. Na and K) arising from sea salt, but weak correlation with its anions, such as m/z 35⁻
and 37⁻ chloride isotopes. A sea salt contribution can thus be ruled out. Its negative spectrum significantly correlates with nitrate,
sulphate, and dust particles. Besides sodium rich dust aged sea salt may be an appropriate classification. Class 3 is named "aged
soot", since it has significant correlation with soot particles, especially diesel soot, and a prominent sulphate signal. This class
has an EC pattern with m/z 12n $C_n^+$, similar to those in the reference spectra for soot particles (Fig. 6) as well as the reference
spectra for PSL particles (Fig. S9). The patterns at m/z 27 $C_2H_3^+$ and 28 $CO^+$, m/z 36 $C_3^+$ and 39 $C_3H_3^+$ as well as the m/z 24⁻, 25⁻
and 26⁻ with higher m/z 26⁻, indicate an OC contribution. This is supported by the correlations especially with PSL particles but
also several other organic compounds, suggesting that this class of particles contains organic species. Class 6 is dominated by
calcium (Ca) and sulphate with characteristic calcium signature peaks at m/z 40 $Ca^+$ and 56 $CaO^-$, also found in the spectra for
dust particles (Fig. 7, Fig. S10, and S11). M/z $40^+$ and $56^+$ may also contain 40 $C_2O^+$ and 56 $Fe/C_4H_8^+$ fragments, respectively.
Class 1 contains almost all fragments observed in other classes, and is thus named "more aged /mixed particles". As shown in
Fig. 9, class 6 is consequently correlated with almost all of the reference spectra (both positive and negative ones).
In order to further interpret the field data, we also classified the ambient mass spectra only based on correlation with 17
selected laboratory-based reference spectra (10 positive + 7 negative spectra) listed in Table S3. This approach resulted in 13
particle types, 7 more than were distinguished by Fuzzy clustering. It should be mentioned that at the beginning we were able to
identify all but the Ca rich particle class resulting from Fuzzy clustering, since initially we did not have a reference for this type.
We therefore used class 6 as an additional reference spectrum for this type of particles, which is among one of the 13 types.
Initially, using a Pearson's correlation coefficient r of $\geq$ 0.6 as threshold for classification resulted in 21 main types of particles
(here we use "types" instead of "classes" in order to differentiate these two classification methods), with particle number
fractions >1%. The corresponding histogram of these 21 particle types is shown Fig. S13. These 21 types were then manually
aggregated after observing their spectra and reduced to 13. Similar as the Fuzzy class number, the resulting number of
characteristic types also strongly depends on the expert experience to identify them (please refer to the details of reference
spectra-oriented grouping procedure in the supplementary information) Their corresponding spectra are shown in Fig. 10. All the
types above the dashed line (A to I) exhibit more prominent secondary inorganic signatures (m/z 97 $HSO_4^-$) and higher number
fractions than the ones below the dashed line. Although particle types A-I all exhibit a more prominent sulphate pattern with m/z
80 and 97 than nitrate pattern with m/z 46 and 62, they are higher correlated with the mixture of nitrate and sulphate than either
of them. Therefore, we assign the corresponding types to nitrate and sulphate. All the types in the lower panels (J to M) have
significant correlation with arable soil dust in the negative spectra, which have organic signatures, e.g. m/z 24⁻, 25⁻, and 26⁻, as
well as some mineral signatures like m/z 119⁻. Compared with the negative spectra, the positive spectra are more characteristic,
which was also observed in the Fuzzy results. Type A, B, C, D, and E are comparable with Fuzzy class 5, 4, 2, 6, and 3,
respectively (the correlation coefficients are 0.89 for type A and class 5, 0.95 for type B and class 4, 0.84 for type C and class 2,
0.76 for type D and class 6, and 0.81 for type E and class 3). Types F to I are more similar to aged/mixed particles, with more
fragments compared to types A to E. Type H is comparable with Fuzzy class 1. ~10% of the particles cannot be grouped into any
type due to spectrum-to-spectrum peak shifts. As shown in the spectra in both Fig. 9 and 10, all organic species were internally
mixed with inorganic species.
This reference spectra-based classification can also be used for identification of particles with low number fractions among
the huge amount of ambient data, and for selection of particles containing particular species e.g. for which the instrument has a
lower sensitivity. This can be achieved by e.g. excluding peaks with high signal such as m/z 39 $K/C_3H_3^+$, or selecting a certain
particle size range, or mass range. As an example, 55 lead containing particles (Pb, with isotopes at m/z 206, 207, and 208)



(details are given in the supplementary information) were identified among the 7314 ambient aerosol particles. The resulting
spectra of particle classes/types in one field study can also be used as reference for other studies. More applications of these
procedures for field data interpretation will be presented in an upcoming paper.

447       In short, Fuzzy and reference spectra-based classifications have some comparable results with high correlations (r: 0.76–0.95)

and also have different advantages: Fuzzy classification can identify special ambient particle types without any existing reference
if they have a significant abundance and signal strength, while reference spectra-based methods can identify target particle types
even with little abundance. They are complementary to some extend and thus their combination has the potential to improve
interpretation of field data.
**4 Conclusions**
In this study, the overall detection efficiency (ODE) of LAAPTOF was determined to range from ~$(0.01 \pm 0.01)$% to ~$(6.57 \pm$
$2.38)$% for polystyrene latex (PSL), ammonium nitrate ($NH_4NO_3$) and sodium chloride (NaCl) particles in the size range between
200 and 2000 nm. This is a relative good detection efficiency compared to earlier versions of the instruments especially when
considering the good reproducibility and stability even during field measurements. A comparison to other single particle mass
spectrometers is subject of another study and will be discussed in a separate publication. In any case matrix effects from aerosol
particles (e.g. size, morphology and optical property) and certain instrument influences (e.g. aerodynamic lens, detection system)
and their interaction must be taken into account to evaluate the LAAPTOF performance.

460       In order to facilitate the interpretation of single particle mass spectra from field measurements, we have measured various

well defined atmospherically relevant aerosol particles in the laboratory and provide here laboratory-based reference spectra for
aerosol particles of different complexity with comprehensive spectral information about the components (such as organic
compounds, elemental carbon, sulphate, nitrate, ammonium, chloride, mineral compounds, metals, etc. as commonly observed in
atmospheric aerosol particles). Our results show that the interpretation of spectra from unknown particle types is significantly
supported by using known mass spectral patterns like signature peaks for ammonium, nitrate, sulphate, and organic compounds
as well as typical peak ratios for e.g. potassium, silicon, and chlorides. Spectra for internally mixed particles may show new
clusters of ions, rather than simply a combination of the ions from single component particles. This may be a complication for
data interpretation which can be overcome if suitable reference spectra for correspondingly mixed particles are available.
Organic compounds generally have some ions in common but exhibit variations depending on the compound. Several peaks can
originate from different fragments, m/z $26^-$ and $42^-$ could be $CN^-$ and $CNO^-$ and/or $C_2H_2^-$ and $C_2H_2O^-$, m/z $39^+$ and $41^+$ could e.g.
originate from $K^+$ isotopes or organic fragments, and organic matter can also be ionized to form the typical elemental carbon
pattern with $C_n^{+/-}$ ions. Hence the interpretation is not always unambiguously possible for such particles but may require
additional information (e.g. size, additional marker peaks, or even higher resolution spectra) or comparison to data from other
instruments like on-line aerosol mass spectrometers (e.g. AMS) or chemical ionization mass spectrometers (e.g. FIGAERO-
CIMS).

476       A set of mass spectra obtained in one day of field measurements was used for particle type classification based on Fuzzy

clustering and on the new reference spectra presented in this work. The corresponding 7314 spectra were clustered by a Fuzzy c-
means algorithm, resulting in six different similarity classes. The independent classification of the ambient mass spectra based on
17 selected reference spectra resulted in 13 different particle types which included those classes obtained by Fuzzy clustering.
Compared with the reference spectra, we found that each class has a sulphate signature at m/z 80 $SO_3^-$ and 97 $HSO_4^-$, a nitrate



signature at m/z 46 $NO_2^-$ and 62 $NO_3^-$, an organic compound signature at m/z 24 $C_2^-$, 25 $C_2H^-$ and 26 $C_2H_2/CN^-$ and a nitrogen-
containing organic signature at m/z 26 $CN^-$ and 42 $CNO^-$. Furthermore, we have performed target-oriented classification by using
a selected reference spectrum, which demonstrates the possibility to identify particles with low number fraction among the huge
amount of ambient data, e.g. lead-containing particles.
We conclude that the reference spectra presented in this paper are useful for interpretation of field measurements and for
understanding the impact of mixing on typical mass spectral signatures. Furthermore, the reference spectra should be useful for
interpretation of data obtained by other LAAPTOF versions or other single particle mass spectrometers using a similar ionization
method and comparable mass spectrometers. For future experiments using the LAAPTOF, systematic studies on its sensitivity to
different species, distinguishing the organic and inorganic contribution to the same peak in the spectra, and investigating peak
ratios are still required.
**Data availability**
The reference spectra are available upon request from the authors and will be made available in electronic format via the
EUROCHAMP-2020 data base (www.eurochamp.org).
**Author contributions**
X.S. characterised the LAAPTOF, measured all the particles samples, did the data analysis, produced all figures, and wrote the
manuscript. R.R. helped to characterise the LAAPTOF and to measure some of the particle samples. C.M. provided technical and
scientific support for characterising the LAAPTOF as well as data analysis, and for interpretation and discussion of the results.
WH provided scientific support for interpretation and discussion of the results. T.L. gave general advices and comments for this
paper. H.S. provided technical and scientific support for characterising the LAAPTOF, as well as suggestions for the data
analysis, interpretation and discussion. All authors contributed to the final text.
**Competing interests**
The authors declare that they have no conflicts of interest.
**Acknowledgements**
The authors gratefully thank the AIDA staff at KIT for helpful discussions and technical support, and the China Scholarship
Council (CSC) for financial support of Xiaoli Shen and Wei Huang. Special thanks go to Thea Schiebel, Kristina Höhler, and
Ottmar Möhler for discussions about the soil dust samples, to Isabelle Steinke for discussions regarding the plant samples, to
Robert Wagner for comments on the sea salt samples, to Konrad Kandler for providing the Morocco desert dust samples, to Roger
Funk and Thomas Hill for providing the soil dust samples, to Elena Gorokhova and Matt Salter for providing the sea salt with
skeletonema marinoi culture, and to Aeromegt GmbH for discussions about the LAAPTOF performance and analysis software.



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





**Table 1: Summary of mass spectral patterns**

| Species | Signature peaks in positive spectra | Signature peaks in negative spectra | Typical Peak Ratios histogram $x_0$ (width)* |
|---|---|---|---|
| potassium | 39 $K^+$, 41 $K^+$ | | (I39+I40):I41=~**13.5**(0.9) |
| calcium | 40 $Ca^+$, 56 $CaO^+$ | | |
| aluminium | 27 $Al^+$ | 43 $AlO^-$, 59 $AlO_2^-$ | |
| silicon | 28 $Si^+$, 44 $SiO^+$ | 60 $SiO_2^-$, 76 $SiO_3^-$, 77 $HSiO_3^-$ | (I76+I77):I60=~**1.0** (0.33) |
| silicon & aluminium | 27 $Al^+$, 28 $Si^+$, 44 $SiO^+$ | 43 $AlO^-$, 59 $AlO_2^-$, 60 $SiO_2^-$, 76 $SiO_3^-$, 77 $HSiO_3^-$, 119 $AlSiO_4^-$, 179 $AlSiO_4.SiO_2^-$ | |
| ammonium | 18 $NH_4/H_2O^+$, 30 $NO^+$ | | |
| nitrate | 30 $NO^+$ | 46 $NO_2^-$, 62 $NO_3^-$ | |
| sulphate | | 32 $S^-$, 48 $SO^-$, 64 $SO_2^-$, 80 $SO_3^-$, 81 $HSO_3^-$, 96 $SO_4^-$, 97 $HSO_4^-$, | |
| chloride | | 35 $Cl^-$, 37 $Cl^-$ | (I35+I36):I37=~**3.2**(0.9) |
| elemental carbon | $12_n C_n^+$ | $12_n C_n^-$ | |
| organics | | 24 $C_2^-$, 25 $C_2H^-$, 26 $C_2H_2/CN^-$ | |
| organic acids | | 45 $COOH^-$, 59 $CH_2COOH^-$, 71 $CCH_2COOH^-$, 73 $C_2H_4COOH^-$, 85 $C_3H_4COOH^-$, 99 $C_4H_6COOH^-$, 117 $(CO)_3OOH^-$ | |
| nitrogen-containing organics | | 26 $CN^-$, 42 $CNO^-$ | |
| unsaturated organics | | 25 $C_2H^-$, 26 $C_2H_2^-$ unknown fragments 49- and 73- | |
| aromatic compounds | 77 $C_6H_5^+$, 91 $C_7H_7^+$, 103 $C_8H_7^+$/105 $C_8H_9^+$, 115 $C_9H_7^+$ | 25 $C_2H^-$, 26 $C_2H_2^-$ unknown fragments 49- and 73- | |


Note:
*We have made histograms for the three typical peak ratios, respectively (ref. Fig. S3). Histogram $x_0$ is the expected value that indicates the
position of the peak resulting from Gaussian fit, and the width is the corresponding standard deviation. I is short for the intensity of the
corresponding peak in LAAPTOF spectra; typical peak ratios for potassium and chloride are based on pure and mixed salt that containing K
and Cl; typical peak ratios for silicon are based on pure $SiO_2$.





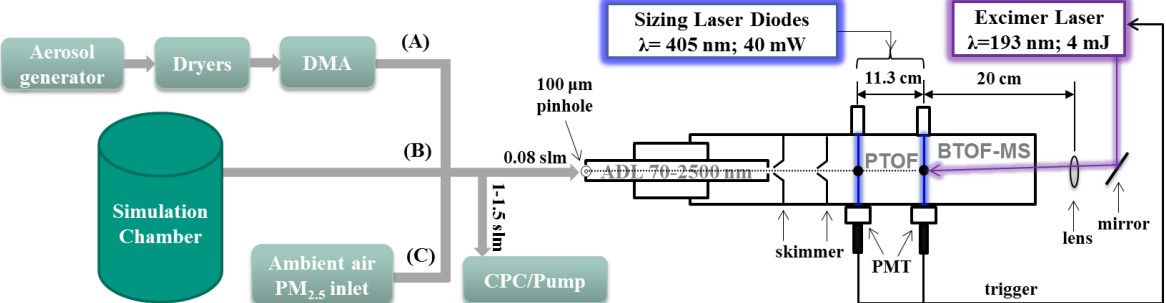


**Figure 1: Schematic of the LAAPTOF instrument and three different experimental setups for measuring standard samples: setup (A)**
**e.g. PSL, NH₄NO₃ and K₂SO₄ particles, were generated from a nebulizer, passed through two dryers, size-selected by a differential**
**mobility analyzer (DMA), and then measured by LAAPTOF; setup (B) was used for samples generated in or dispersed into the AIDA**
**chamber (~84.5 m³) or samples dispersed into a stainless steel cylinder (~0.18 m³); setup (C) was used for measuring ambient aerosols**
**in field campaigns. In addition, some particles, e.g. mineral dust, were sampled directly from the headspace of their reservoirs.**





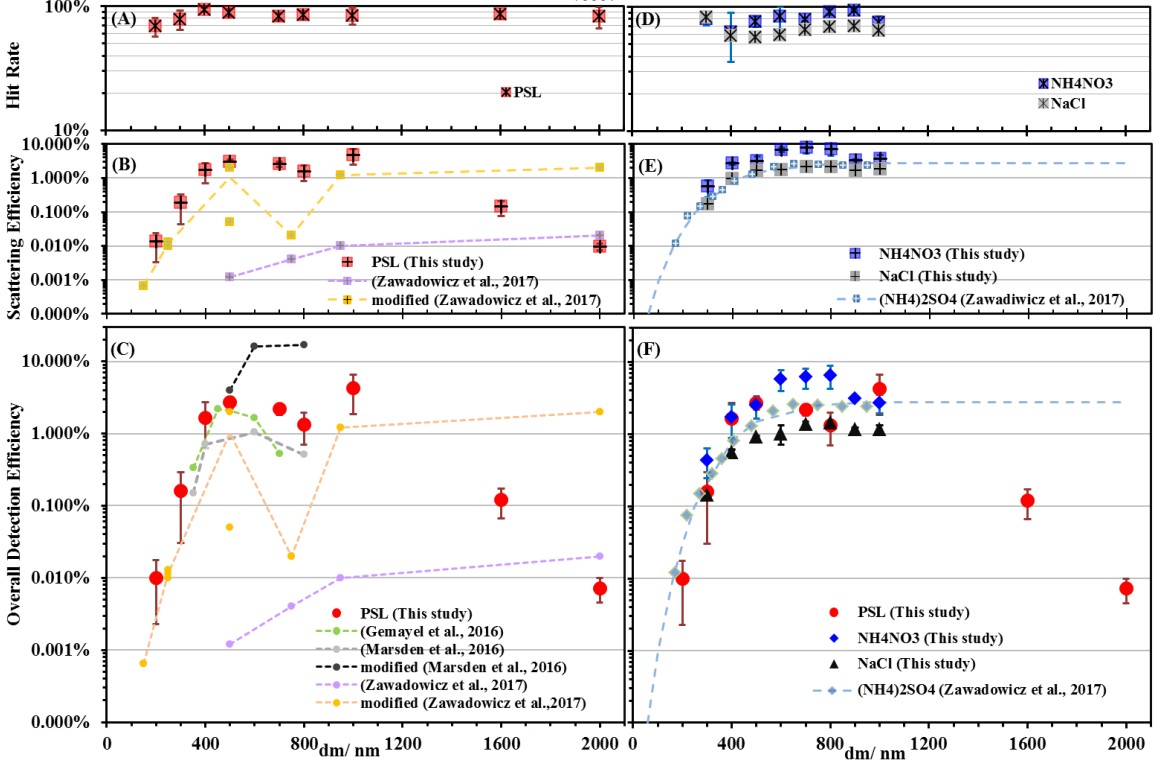


**Figure 2: Hit rate (HR, panel A and D), scattering efficiency (SE, panel B and E), and overall detection efficiency (ODE, panel C and F) for PSL, ammonium nitrate (NH₄NO₃) and sodium chloride (NaCl) salt particles as a function of mobility diameter, $d_m$. Aerosol particles in this study were generated from a nebulizer and size-selected by DMA. In panel (B) and (E), optical counting efficiencies (OCE) for PSL and ammonium sulphate ((NH₄)₂SO₄) at the detection beam from the study by Zawadowlcz et al. (2017), corresponding to the SE defined in this study, are plotted for comparison. In panel (C) and (F), ODE for PSL and salt particles from other studies (Gemayel et al., 2016; Marsden et al., 2016; Zawadowicz et al., 2017) are plotted for comparison. In this figure, dashed lines are used only for guiding the eyes.**





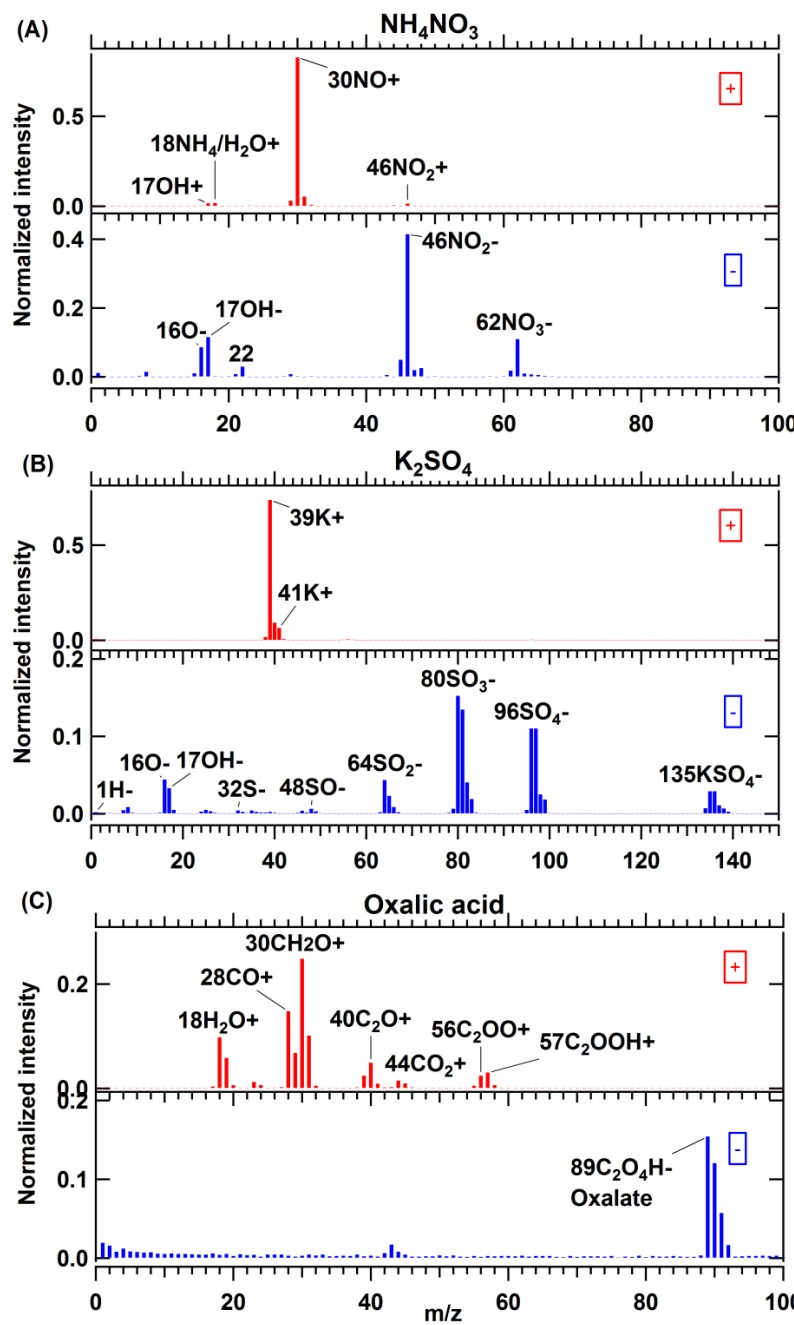

682    **Figure 3: Average mass spectra for pure compound aerosol particles: (A) NH₄NO₃ (dva=1160 nm), 497 single spectra averaged, (B)**
683    **K₂SO₄ (dva=1465 nm), 300 single spectra averaged, and (C) oxalic acid particles (dva=1081 nm ), 736 single spectra averaged.**





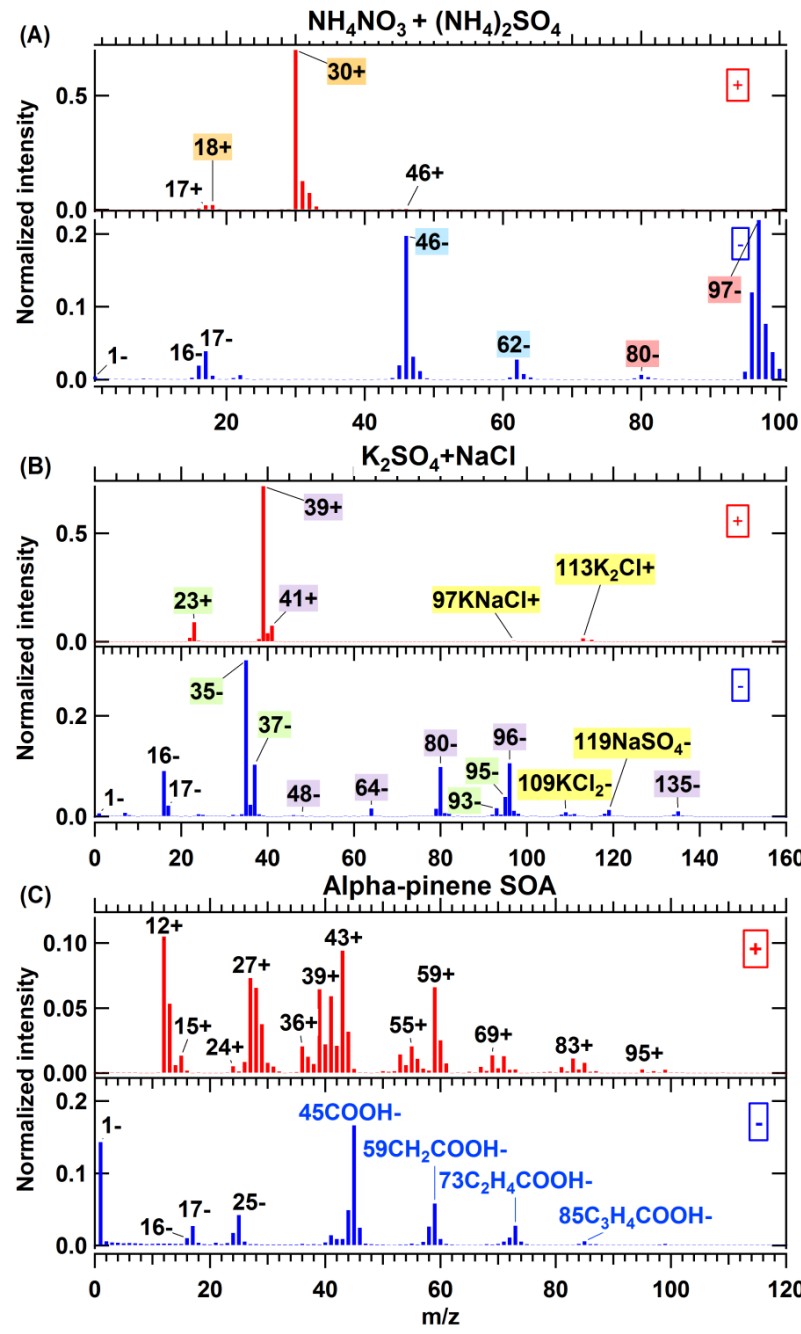

684

**Figure 4: Average mass spectra for particles of internal mixtures of (A) NH₄NO₃ and (NH₄)₂SO₄, (dᵥₐ= 1102 nm), 454 single spectra averaged and (B) NaCl and K₂SO₄, (dᵥₐ= 1375 nm), 259 single spectra averaged as well and (C) secondary organic aerosol (SOA) particles from α-pinene ozonolysis, which was performed in the APC chamber, then the resulting particles were transferred into the AIDA chamber at 263 K and 95% RH, dᵥₐ= 505 nm, 1938 single spectra averaged. In panel (A), red, blue and orange labels represent fragments of sulphate, nitrate and ammonium, respectively. In panel (B), green and purple labels represent fragments from NaCl and K₂SO₄ components (see section 3.2.1) in the mixed particles, respectively; yellow labels represent the fragments only in the internal mixture of NaCl and K₂SO₄. In panel (C), labels with blue text represent fragments of organic acids.**




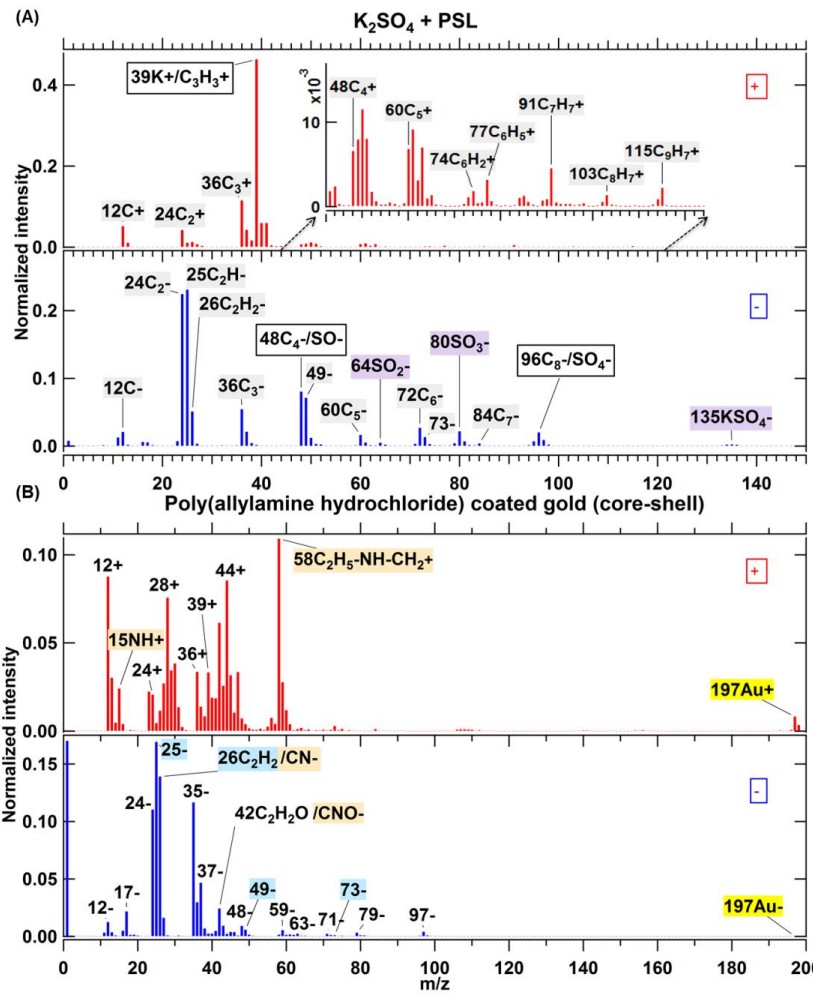

692

**Figure 5: Average mass spectra for core-shell particles of (A) PSL coated with K₂SO₄, d_va= 805 nm, 609 single spectra averaged, and (B) poly(allylamine hydrochloride) coated gold (Au) particles with geometric 300 nm gold core and 50 nm thick organic shell, 417 single spectra averaged. In panel (A), grey and purple labels represent the fragments arising from pure PSL and pure K₂SO₄ components, respectively; box labels represent the fragments with contributions from core and shell compounds. In panel (B) orange and blue labels represent the fragments arising from nitrogen-containing and unsaturated organic compounds, respectively, and yellow labels represent gold.**





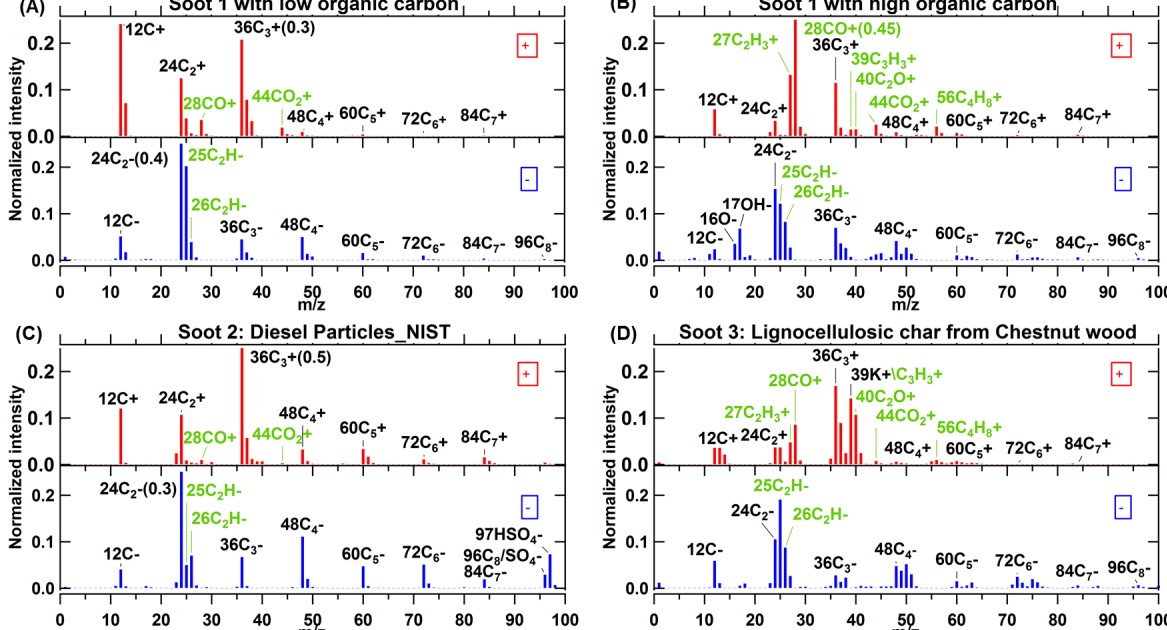

**Figure 6:** Average mass spectra for soot particles with (A) high elemental carbon (EC), low organic carbon (OC) content and (B) low EC and high OC from combustion of propane in a soot generator and transferred to a stainless steel cylinder of ~0.2 m³ volume, as well as soot particles of (C) diesel particles (NIST) and (D) lignocellulosic char from Chestnut wood. In panel (A) and (C), the numbers in brackets beside peak 36+ and 24- are the exact intensity values for them. The OC signatures are labeled in green. The numbers of spectra averaged for each spectrum are 617 (A), 347 (B), 533 (C) and 390 (D).





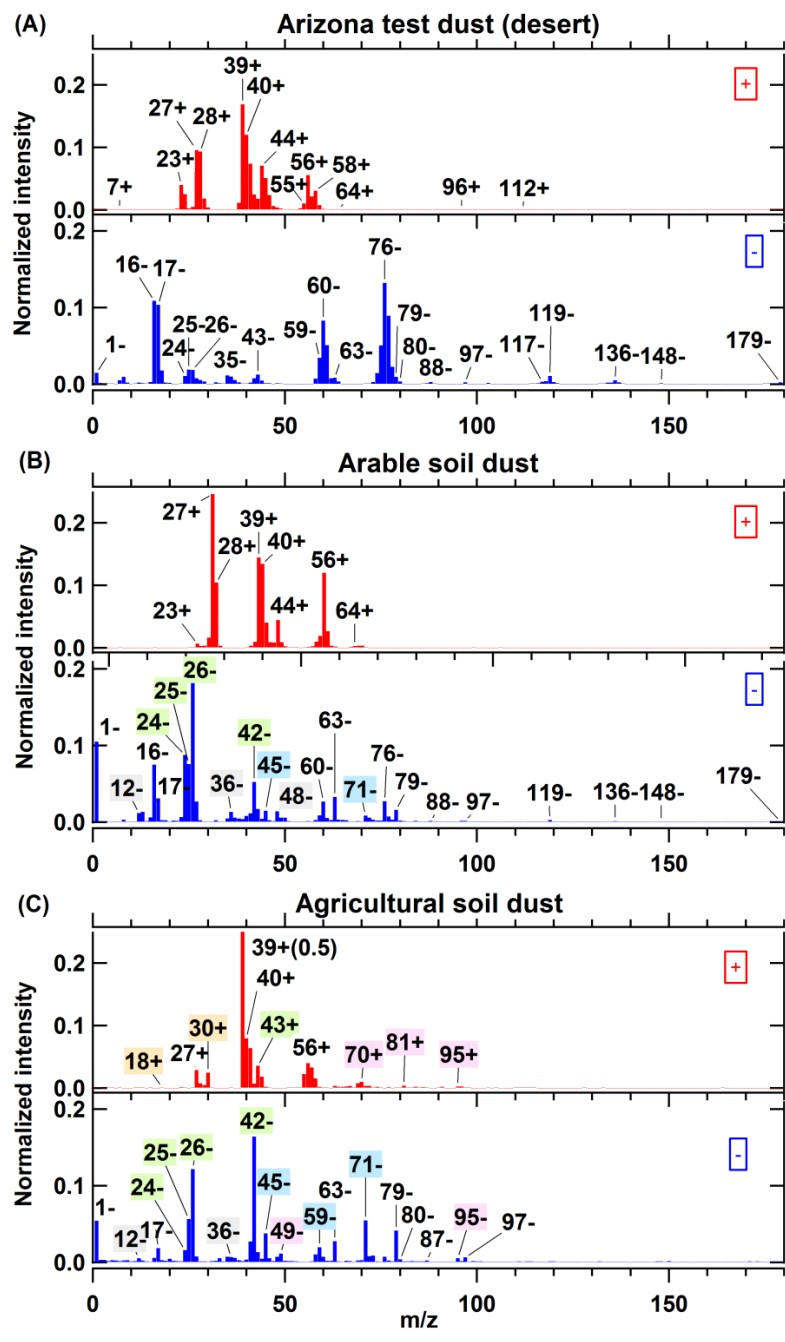

705

**Figure 7:** Average mass spectra for particles of complex mixtures: (A) Arizona test dust (desert dust), directly sampled into the LAAPTOF from a shaked bottle (B) arable soil dust, collected from Gottesgabe in Germany, was dispersed by a rotating brush generator and injected via cyclones into the AIDA chamber at 256 K and 80% RH, and (C) agricultural soil dust, collected from harvesting machines after rye and wheat harvest, were generated by using the same method as (B). For panel (B) and (C), fragments labelled in green represent more intensive organic signatures in soil dust particles; grey labels represent EC patterns; blue labels represent organic acids; orange labels represent ammonium salts; red labels represent unknown fragments. The numbers of spectra averaged for each spectrum are 261 (A), 583 (B), and 286 (C).





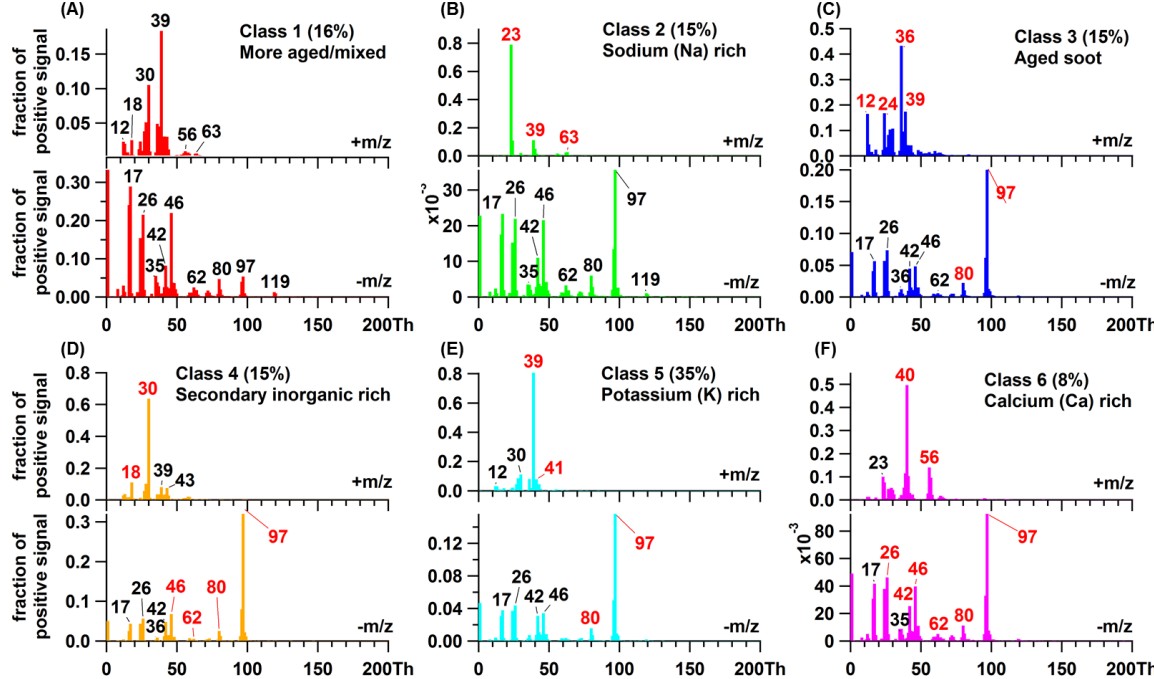

713

**Figure 8: Mass spectra for six classes of particles measured on July 29th, 2016 during the field campaign TRAM01, based on**
**classification according to Fuzzy c-means algorithm. The percentage in each pair of spectra (A to F) gives us information about the**
**similarity of the total aerosols to different classes. The red tags represent the signatures for each typical class, but there is no red tag in**
**spectra B, since this class is more aged particles that containing signatures for different classes. Mean particle size: $d_{va}$ (676±165) nm.**



718

Figure 9: Correlation between Fuzzy classification results (6 classes, C1 to C6) and laboratory-based reference spectra. Panel (A) and
(B) are the correlation results for the positive and negative spectra, respectively. AN is short for ammonium nitrate, PS-potassium
sulphate, SC-sodium chloride, PinA-pinic acid, Pino-pinonic acid, HA-humid acid, OA-oxalic acid, ATD-Arizona test dust, CD-Cairo
dust, MD-Morroco dust, UD-urban dust, SDGe01 and SDPA01-soil dusts sampled at two sites from Germany, SDAr08- soil dust from
Argentina, SDWY01-soil dust from Wyoming in USA, ASD-agricultural soil dust, ECS-EC rich soot1, OCS-OC rich soot1, DS- diesel
soot, BS- biomass burning soot, which is the lignocellulosic char from Chestnut wood, SS-pure sea salt, SSO-sea salt with organics.

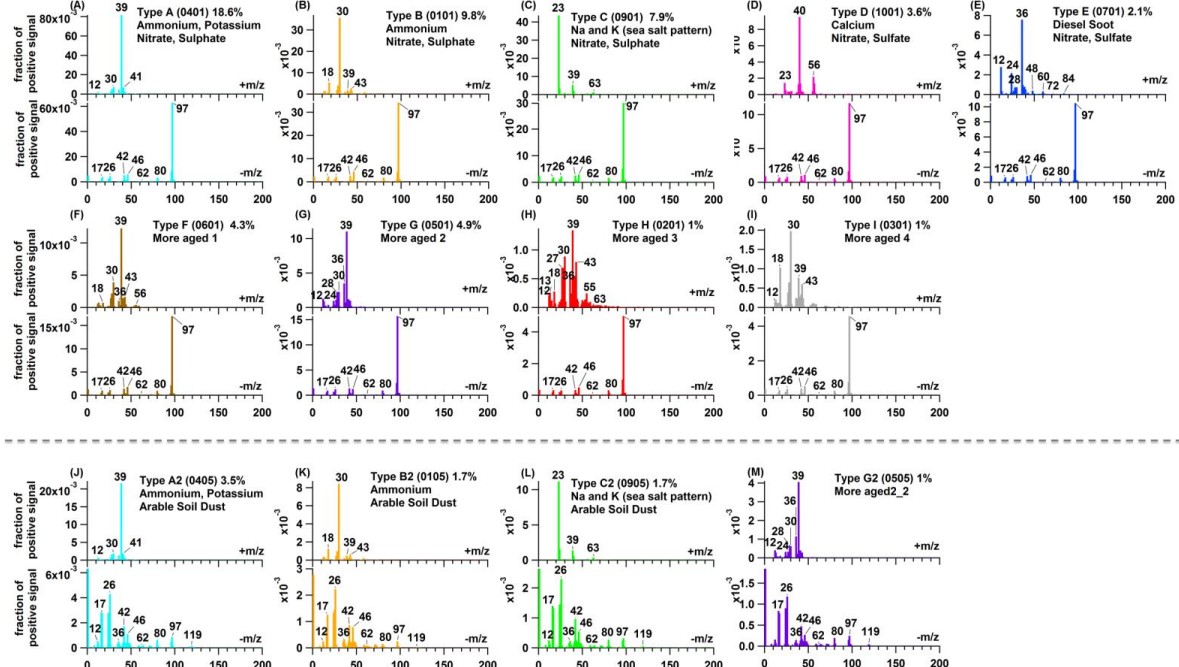


**Figure 10: Mass spectra for 13 different types of particles measured on July 29th, 2016 during the field campaign TRAM01, based on**
**the classification according to laboratory-based reference spectra. The 4-digits codes in the brackets represent particle types (c.f. Table**
**S3). The % values are the particle number fractions. For panel A to E and J to L, there are two lines for the names, the first and**
**second lines correspond to the highly correlated positive and negative references, respectively.**