# Peer review of "Laser ablation aerosol particle time-of-flight mass spectrometer (LAAPTOF): Performance, reference spectra and classification of atmospheric samples"

_Atmospheric Measurement Techniques, 2017_

## Referee Comment (RC1) · Anonymous Referee #2 · 20 Dec 2017

It is my pleasure to provide a review for this manuscript. The paper describes a comprehensive set of test measurements conducted with a new single particle mass spectrometer, the LAAPTOF. The study provides a good overview over the instrument's core capabilities and performance. As there is limited information on the LAAPTOF in existing literature, this study is a much-needed reference for current and future LAAPTOF users, and for the aerosol mass spectrometry community in general. In terms of content, there is a wealth of information in this paper, with little to add or change; except some of the choices in the field data analysis should be explained better. In terms

of organization and readability, there is room for improvement: The laboratory measurements presented in the paper were done with several different setups and sample preparation techniques, and, for interpretation, the samples are grouped according to the mixing state of the aerosol. Some of the measurements were size-selective whereas others were not, but for those that weren't, sizes are still reported. Currently, the information which aerosol type was generated how, sampled with what setup, and assigned to which group, is contained in the flow text of section 2.2, and in Table S1 and its footnotes. Both of these are difficult to refer back to quickly, and it is hard for the reader to keep it all straight. The presentation of experimental setups (Figure 1) is also a little confusing, as different laboratory setups are conflated with the field setup (making it look like one experimental setup even though it is not) and simple schematics are combined with a detailed functional sketch of the LAAPTOF instrument.

I recommend final publication after consideration of the following suggestions and comments:

1. The correlation method (a consistent name should be used for this method throughout the paper) should be explained better. Why were positive and negative spectra treated separately (10 and 7, respectively)? Aren't the positive-negative pairs intimately connected, coming from one particle? How were they re-combined to yield 13 particle types (Figure 10)? The caption of Figure S13 contains hints, but is extremely hard to understand. Also, referring to lines 371 – 372 (and the procedure description in the supplement): there are ways to determine the appropriate number of clusters aside from "experience of the operating scientist" – see for example Hinz et al. (1999). Is there a specific reason these were not used?

2. How was the particle size information obtained and handled? According Section 3.2, line 246, a dm of 800 nm was chosen (presumably only for the samples passing through the DMA). In Table S1, this size only appears for PSL, and other sizes (dva) are given for all samples. One of the footnotes mentions a Gaussian fitting, but this should be described more clearly in the "methods" section.

3. Organization: There are three experiments here: 1. a rigorous determination of ODE for some aerosol types, 2. a comprehensive collection of reference spectra, 3. a field study with two different data analysis methods. For the sake of readability, the three should be treated separately as much as possible. Here are some suggestions to improve on clarity:

a) Move table S1 into the main manuscript. In the footnotes, make sure that the explanations for the generation methods are easy to spot – perhaps a list, rather than wrapped into a sentence.

b) The three (or four, including sampling from reservoir headspaces?) experimental setups should be defined in Section 2 (separately from the generation methods) and referred back to throughout the paper. Generation methods A, B1, B2, and S should also be defined in the text in Section 2, and referred back to.

c) My suggestion for Figure 1 would be to only show setup A, as there is no information in setups B and C that cannot be described easily in the text. Setup A should then be shown in more detail (flow rates, flow splitting between LAAPTOF inlet and CPC, etc.). This would make it easier for the reader to mentally separate the ODE experiments from the reference spectra experiments. Also, in the Figure, a box or some other designation should be put around the actual LAAPTOF instrument. One could also show setup B in a separate figure, with more detail on the particle generation (cyclones, APC chamber, CAST, etc.), but I believe it is not necessary, as there are descriptions and references in section 2.2.

d) Section 3.1. should be reorganized. Lines 167 – 174 in Section 3.1.1 should be part of Section 2 (Methods). Section 3.1.2 is currently a mix of discussion of ODE measurement results, interpretation of features in the reference spectra, and literature review. While the factors mentioned in lines 223 - 239 may influence ODE, their impact is not put into quantitative relation to the presented measurement results. The discussion would be more appropriately placed in the introduction (some of it, such as the

40 mW laser setting, in Section 2.1). I would also argue that lines 211 – 221 could be discussed in the context of the reference spectra (3.2); at least, though, there should be several references to Section 3.2.

f) The data analysis methods should be presented in a separate subsection of Section 2. The sections headed "Procedures for ..." and "Seeking..." in the supplement (which could be very useful for other LAAPTOF users), should be proofread (informal expressions), designated ( "Procedure 1", "Procedure 2", or something similar) and referred to in Section 2.

4. Presentation of Figures: Figure 2: Panel A is very important, but hard to read. Consider putting the y-axis on a linear scale and repeating the x-axis on the top of the graph. Figures 4-7: It is hard for the reader to go back between label shadings in the various panels and description of the shadings in the caption. A legend for the shading in or next to each panel would be much more convenient. What are the dva values in Figure 4 referring to exactly? Figures 4 C (+), and 7: It would be nice to have a few more peak labels (ions, not just m/z values). Figure 9: As much as possible, avoid extra abbreviations, they are not reader-friendly. For example, there is no reason that the axis tick label "SC" could not be "NaCl". Figure 10: In the current layout, this Figure is too small.

Other comments:

5. lines 82-86: There is another recent paper describing LAAPTOF spectra by Wonaschuetz et al. (2017, Journal of Aerosol Sience 113, 242 - 249)

6. line 124: Can provide a short description of what exactly is integrated in "stick integration"?

7. line 130-131, explanation of percentages: Is this quantity ever used in the paper? If not, this explanation could be omitted.

8. line 141: Consider "dissolved/suspended" (PSL spheres don't dissolve)

9. line 149: Can you add a very short description of the "different ways"?

10. line 161: How long was the LAAPTOF deployed?

11. line 173: In the samples generated by nebulizing salts, followed by size selection in the DMA, how was the possibility of multiply charged particles handled?

12. line 200: Can you provide a reference?

13. line 203: It seems that it would be easy to provide a quick comparison of Mie scattering efficiencies at the particle sizes/laser wavelength relevant to this measurement to validate the M-shape.

14. line 234: Was this controlled for in some way in the test measurements?

15. line 271: This is confusing. Which ratio is 1?

16. line 285: Can you provide a reference?

Minor corrections/typos:

17. line 296: "can be found".

18. line 305: "are weaker"

19. line 328: "panel A"

20. line 312: "are observed"

21. line 332 – 337: This is a very long sentence. Consider this construction: "In spectra (A), we also observed the following peaks: ...."

22. line 339: consider: "Most of the ... fragments of soil dust are similar to those of desert dust..."

23. line 348: Start the sentence with "The" and remove the "s" in "soil dust(s)" from Argentina".

24. lines 454 – 456: "e.g." is in a strange place in the sentence. Perhaps: "...from different fragments: for example, m/z 26 ...."

25. line 442: "In any case" seems to be a needless expression.

26. line 467: consider replacing "among the huge amount of ambient data" by "in the ambient aerosol"

---

## Referee Comment (RC2) · Anonymous Referee #1 · 4 Jan 2018

The authors present performance data and reference spectra from the commercially available LAAP-TOF single-particle mass spectrometer. The manuscript is of a technical nature and provides a characterisation of the instrument that can be used as a platform for future scientific investigation with this instrument model. In addition, an alternative particle classification method is presented and compared to the fuzzy clustering algorithm provided by the instrument manufacturer. Although performance data in terms of detection efficiencies have previously been reported in the literature, the instrument has been the subject of continual development; consequently data for this

particular example of the instrument design is relevant. The reference spectra presented in this manuscript are from a wide variety of atmospherically relevant particles types and are likely to be very useful to other users of this instrument platform. I gladly recommend this manuscript for publication in AMT.

General Comments: There are number of details that should be addressed before publication. The manuscript can be difficult to read in places and some streamlining of the introduction would help. Also, the authors should be careful to state which conclusions are drawn from which measurements, particularly when the detection efficiency with respect to particle size is discussed. Finally, the comparison of the clustering results with reference spectra is very informative and should be given more prominence in the abstract and conclusions.

Specific Comments:

Pg1, ln16. Only the PSL was measured at the size range stated.

Pg1, ln21. It is not clear what is meant here. Clarify that you are using two methods to analyse the ambient data?

Pg1, ln31. This paragraph could be streamlined.

Pg2, ln53. Not sure this review of SPMS history is necessary.

Pg2, ln67. There is an instrument commercial available from Hexin instruments, China.

Pg2, ln68. How is the beam spot diameter known? Measured or referenced?

Pg2, ln73. In Figure2, the ODE of 1% appears to be achieved BEFORE instrument modification. Which is correct, the figure or the text?

Pg3, ln82. Zawadowicz (2017) has a beam spot size of 100$\mu$m after focussing.

Pg3, ln84. Improved optical counting efficiency (2-3 order of magnitude) with respect to what?

Pg3, ln94. This paragraph should be in the method section.

Pg4, ln119. Is there a reference for the stated transmission efficiency?

Pg4, ln123. Was the beam diameter measured or referenced? This is important for the stated power density.

Pg4, ln124. Power density is measured in W/cm3. Are the authors referring to 'Intensity'?

Pg4, ln. Differences in peak position mainly occur due to differences in kinetic energy of the ions produced. A suitable reference should be given.

Pg5, ln163. It is not clear how particles are sampled from the 'Head space of their reservoirs'

Pg5, ln 165. This paragraph is a repeat of the information offered in the introduction.

Pg5, ln175. The definition 'scattering efficiency' is ambiguous as it can be confused with the optical properties of the particle rather than the efficiency of the system. Suggest 'particle detection efficiency' or similar.

Pg6, ln 205. The authors should be clear if they are discussing the LAAP-TOF detection efficiency specifically. It is possible to detect particle smaller than 200nm with some systems.

Pg7, ln216. A size range should be given for this statement. The scattering efficiency of non-spherical particles $> 1 \mu$m is not reported in this manuscript.

Pg7, ln217. This paragraph is confusing and should be re-written. It should be clear if the authors are discussing the optical properties associated with absorption and hit-rate or the optical properties associated with particle detection. Please us the definitions already given in the manuscript.

Pg7, ln233. The information in this paragraph was already given in the introduction.

Pg8, ln252. Can the authors offer any data to support the conclusion that mass spectral signatures increase with particle size?

Pg8, ln275. Are the authors referring to detector ringing? Detector ringing in MCPs is caused by pulse reflections and therefore should not be counted as ion signal.

Pg12, ln437. Please explain what 'spectrum-to-spectrum peak shifts' is referring to and how that impacts the assignment of particle type.

Pg13, ln447. This paragraph contains some nice conclusions about particle classification and should be given more prominence in the manuscript.

Pg13, ln454. Only PSL was measured in that size range.

Typos/Technical:

Pg2, ln 51. Sentence structure.

Pg 2, ln 55. Home-built.

Pg3, ln76. 'Number percentage' should be 'number fraction'?

Pg3, ln104. Remove 'Major'.

Pg3, ln106. Insert 'and' after SOA.

Pg3, ln109. Sentence structure.

Pg5, ln194. Check the spacing after the references.

Pg10, ln360. Poor sentence structure.

---

## Author Comment (AC1) · 2 Feb 2018

**Author comment on "Laser ablation aerosol particle time-of-flight mass spectrometer (LAAPTOF): Performance, reference spectra and classification of atmospheric samples" by Xiaoli Shen et al.**

*We gratefully thank the reviewers for their careful manuscript reading, and their helpful comments to improve the quality of our manuscript. Our point-to-point replies to the individual comments are in italics, marked by R. as follows:*

**Referee #1 comments:** The authors present performance data and reference spectra from the commercially available LAAP-TOF single-particle mass spectrometer. The manuscript is of a technical nature and provides a characterization of the instrument that can be used as a platform for future scientific investigation with this instrument model. In addition, an alternative particle classification method is presented and compared to the fuzzy clustering algorithm provided by the instrument manufacturer. Although performance data in terms of detection efficiencies have previously been reported in the literature, the instrument has been the subject of continual development; consequently data for this particular example of the instrument design is relevant. The reference spectra presented in this manuscript are from a wide variety of atmospherically relevant particles types and are likely to be very useful to other users of this instrument platform. I gladly recommend this manuscript for publication in AMT.

General Comments: There are number of details that should be addressed before publication. The manuscript can be difficult to read in places and some streamlining of the introduction would help. Also, the authors should be careful to state which conclusions are drawn from which measurements, particularly when the detection efficiency with respect to particle size is discussed. Finally, the comparison of the clustering results with reference spectra is very informative and should be given more prominence in the abstract and conclusions.

*R: 1) Thank you for your advice, we have streamlined the introduction section by shortening its 2$^{nd}$ paragraph and review of SPMS history (refer to our reply R3 and R4 to the specific comments no. 3 and no. 4, respectively) as well as moving the 5$^{th}$ paragraph to the method section 2.5 Spectral and size data analysis (refer to the specific comment 10).*

*2) Regarding the conclusions drawn from the measurements particularly when the detection efficiency with respect to particle size is discussed: We have revised the corresponding contents in the manuscript.*

*3) Thank you for your advice. We have given the comparison of the fuzzy clustering results with reference-based classification in the abstract and conclusions more prominence in the revised manuscript. More details can be found in our reply (R25) to the corresponding specific comment no.25*

*Further detailed revisions addressed in the manuscript as well as the supporting information are given in our replies to the specific comments.*

**Specific Comments:**

1. Pg1, ln16. Only the PSL was measured at the size range stated.

*R1: Yes, you are right. We have revised this sentence as following: "The overall detection efficiency (ODE) of the instrument we use was determined to range from ~(0.01 ± 0.01)% to ~(4.23 ± 2.36)% for polystyrene latex (PSL) in the size rage of 200 to 2000 nm, ~(0.44 ± 0.19)% to ~(6.57 ± 2.38)% for ammonium nitrate (NH$_4$NO$_3$), and ~(0.14 ± 0.02)% to ~(1.46 ± 0.08)% sodium chloride (NaCl) particles in the size rage of 300 to 1000 nm."*

2. Pg1, ln21. It is not clear what is meant here. Clarify that you are using two methods to analyse the ambient data?

*R2: We have revised this sentence as following: "An exemplary one-day ambient data set was analysed by both classical Fuzzy clustering and a reference spectra based classification method, generating results with Pearson's correlation coefficients of 0.76 to 0.95."*

3. Pg1, ln31. This paragraph could be streamlined.

*R3: We have streamlined this paragraph as following: "Aerosol particles can contain various components ranging from volatile to refractory species (Pratt and Prather, 2012). The global aerosol mass burden was*

*estimated to consist of 73.6% dust, 16.7% sea salt, 2.8% biogenic secondary organic aerosols (SOA), 2.3% primary organic aerosols (POA), 1.3% sulphate, 1.3% ammonium, 1.2% nitrate, 0.4% black carbon (soot), 0.2% anthropogenic SOA, and 0.2% methane sulphonic acid (Tsigaridis et al., 2006). During the ambient aerosols' lifetime, ranging from hours to a few weeks (Pöschl, 2005), the complexity of their chemical composition usually increases by coagulation, cloud processing, and chemical reactions (Seinfeld and Pandis, 2006). This modifies the particles' mixing state, with both internal (individual particles consisting of mixed compounds, e.g. coating structures) and external mixtures (e.g. mixture of particles consisting of different compounds) (Li et al., 2016). The aforementioned findings underscore the importance of measuring aerosol chemical composition and its changes on short timescales and on a single particle basis, which can be realized by on-line mass spectrometry."*

4. Pg2, ln53. Not sure this review of SPMS history is necessary.

*R4: We have removed this review of SPMS history.*

5. Pg2, ln67. There is an instrument commercial available from Hexin instruments, China.

*R5: We have revised this sentence as "Currently, there are only two commercially available SPMSs, i.e. the Single Particle Aerosol Mass Spectrometer (SPAMS, Hexin Analytical Instrument Co., Ltd., China) (Li et al., 2011; Lin et al., 2017) and the Laser Ablation Aerosol Particles Time-of-Flight mass spectrometer (LAAPTOF, Aeromegt GmbH, Germany)."*

6. Pg2, ln68. How is the beam spot diameter known? Measured or referenced?

*R6: The beam spot diameter is known from references (Marsden et al., 2016; Zawadowicz et al., 2017). We have added these two references.*

7. Pg2, ln73. In Figure2, the ODE of 1% appears to be achieved BEFORE instrument modification. Which is correct, the figure or the text?

*R7: The figure is correct. We have revised the text as "The instrument used by Gemayel et al. (2016) exhibited a maximum ODE of ~2.2% for PSL particle diameters of 450 nm, while ~1% at 600 nm was the peak ODE reported by Marsden et al. (2016) before the instrument modification."*

8. Pg3, ln82. Zawadowicz (2017) has a beam spot size of 100_m after focussing.

*R8: Thank you for pointing out this mistake. We have corrected this in the manuscript.*

9. Pg3, ln84. Improved optical counting efficiency (2-3 order of magnitude) with respect to what?

*R9: The optical counting efficiency was improved with respect to before instrument modification. We have revised this sentence as following: "Zawadowicz et al. (2017) modified the optical path of the laser diodes with a better laser beam of <1 mrad full angle divergence and 100 µm detection beam spot size, and applied light guides to enhance the scattered light collection. This resulted in 2−3 orders of magnitude improvement in optical counting efficiency of incident PSL particles with 500−2000 nm vacuum aerodynamic diameter ($d_{va}$).".*

10. Pg3, ln94. This paragraph should be in the method section.

*R10: We did not move this paragraph since we want to give the reader a short introduction to the SPMS data analysis options. However, we added more detail to the method section 2.5 Spectral and size data analysis.*

11. Pg4, ln119. Is there a reference for the stated transmission efficiency?

*R11: The stated transmission efficiency of ADL can be found via the Aeromegt website (http://www.aeromegt.com/#products?LPL-2.5_details). We have added this link in the manuscript.*

12. Pg4, ln123. Was the beam diameter measured or referenced? This is important for the stated power density.

*R12: The excimer laser beam focus diameter was measured by Ramisetty et al. (2017).*

13. Pg4, ln124. Power density is measured in W/cm3. Are the authors referring to 'Intensity'?

*R13: The laser power density stated here is calculated as the power per beam area at focal point and thus has the unit of "W/cm$^2$" (Zawadowicz et al. (2015).*

14. Pg4, ln. Differences in peak position mainly occur due to differences in kinetic energy of the ions produced. A suitable reference should be given.

*R14: The effect of differences in kinetic energy of the ions produced can typically be compensated in the TOFs with reflectron (Kulkarni et al., 2011). The differences in peak position mentioned here are mainly due to the variance in the position of particle-laser interaction. We have revised the corresponding paragraph as follows:*

*"It should be noted that spectrum-to-spectrum differences in peak positions for the same ion fragments/clusters complicate the mass calibrations. This may be caused by differences in kinetic energy of the ions produced, however this effect is typically compensated in the TOFs with reflectron (Kulkarni et al., 2011). Spectrum-to-spectrum peak shifts, especially in the positive spectra in our study, are mainly because of variance in the position of particle-laser interaction, which cannot be corrected with the existing Aeromegt software or the LAAPTOF instrument (Ramisetty et al., 2017). Details can be found in 'Procedure 1' in the supplementary information".*

*The corresponding revised paragraph in "Procedure 1" in the supplementary information is: "It should be noted that there are spectra-to-spectra peak shifts (~100 ns) due to variance in the position of the interaction of the individual particles with the excimer laser beam, complicating mass calibration. This cannot be corrected with the existing Aeromegt software or the LAAPTOF instrument. It could be avoided by adding/implementing a pulsed extraction, which would store the ions for a certain time and then extract them into the TOF region. Peak shifting is less problematic for the negative spectra than for the positive spectra."*

15. Pg5, ln163. It is not clear how particles are sampled from the 'Head space of their reservoirs'

*R15: The samples were put into a reservoir (e.g. bottle) in amounts that just covered the bottom. Particles were then suspended by shaking the reservoir, and the suspended particles in the reservoir were sampled through a tube to the LAAPTOF inlet. We have revised the corresponding paragraph: "Silica, Hematite, Illite_NX, Arizona test dust, desert and urban dust, black carbon from Chestnut wood (University of Zürich, Switzerland), and diesel soot reference particles from NIST were sampled directly from a reservoir (e.g. bottle) through a tube connecting it with the LAAPTOF after having been suspended by shaking the reservoir (Method S)."*

16. Pg5, ln 165. This paragraph is a repeat of the information offered in the introduction.

*R16: We think this paragraph contains additional information, e.g., location and goal of the measurements, to provide the reader with a minimum background on the field measurements.*

*The information about the field measurement in the introduction was "A one-day example of field data interpretation based on these reference mass spectra will be given in chapter 3.3 and compared to a Fuzzy clustering approach.". This has been rephrased as "An example for field data analysis based on reference spectra as well as Fuzzy c-means clustering will be given in chapter 3.3.".*

17. Pg5, ln175. The definition 'scattering efficiency' is ambiguous as it can be confused with the optical properties of the particle rather than the efficiency of the system. Suggest 'particle detection efficiency' or similar.

*R17: Thank you for your suggestion. However, it might cause misunderstanding if we use "particle detection efficiency", which is used also for the overall detection efficiency of the SPMS, including the detection and ionization regions. Therefore, we will keep "scattering efficiency", but we have added a more clear definition ("the fraction of particles detected by the scattering optics in the detection region of the instrument") to the method section 2.4.*

**Author comment on "Laser ablation aerosol particle time-of-flight mass spectrometer (LAAPTOF): Performance, reference spectra and classification of atmospheric samples" by Xiaoli Shen et al.**

18. Pg6, ln 205. The authors should be clear if they are discussing the LAAP-TOF detection efficiency specifically. It is possible to detect particle smaller than 200nm with some systems.

*R18: Thank you for reminding us. Yes, we are only discussing the LAAPTOF overall detection efficiency. Therefore, we have revised the sentence as "There are various factors that can influence the ODE of LAAPTOF".*

19. Pg7, ln216. A size range should be given for this statement. The scattering efficiency of non-spherical particles > 1m is not reported in this manuscript.

*R19: Yes, you are right. We have revised it as "However, in the size range of 300 to 1000 nm studied here, they don't exhibit Mie resonance and thus don't show an M-like shape in their scattering efficiency."*

20. Pg7, ln217. This paragraph is confusing and should be re-written. It should be clear if the authors are discussing the optical properties associated with absorption and hitrate or the optical properties associated with particle detection. Please us the definitions already given in the manuscript.

*R20: We have re-written this paragraph as following: "Optical properties of the particles have a strong impact on how light is scattered and absorbed, and thus it should be noted that the optical properties do not only influence scattering efficiency, but also absorption and ionization efficiency (or hit rate). As shown in Fig. 2F, ODE for $NH_4NO_3$ is higher than that for NaCl at any size we studied. This is mainly caused by differences in their optical properties of scattering. Relative fresh soot particles scatter only little light due to their black colour and small size (typically ~ 20 nm) of the primary particles forming their agglomerates, and are thus hardly detected by the detection laser. However they are good light absorbers and thus relatively easy to ablate and ionize. The reference spectra of pure $NH_4NO_3$ and $(NH_4)_2SO_4$ particles showed intensive prominent peaks for pure $NH_4NO_3$ particles (refer to Fig. 3A) but only one weak peak of m/z 30 $NO^+$ for pure $(NH_4)_2SO_4$ particles. This indicates that $NH_4NO_3$ is a better absorber than $(NH_4)_2SO_4$, and thus easier to ablate and ionize. For homogeneous mixtures of these two ammonium salts, the sulphate species are ablated and ionized much more easily (refer to section 3.2.2), due to increased UV light absorption by the nitrate component. Some small organic compounds with weak absorption properties are hard to ablate and ionize, e.g. oxalic acid ($C_2H_2O_4$), pinic acid, and cis-pinonic acid. They exhibited much weaker signals (~80% lower) than macromolecular organic compounds in PSL or humic acid particles."*

21. Pg7, ln233. The information in this paragraph was already given in the introduction.

*R21: Yes, you are right. We have shortened this paragraph as "The incident intensity of radiation, which is another parameter that influences the light scattered by particles (as well as background signal caused by stray light), is related to power and beam dimensions of the detection laser. Corresponding instrument modifications were done by Marsden et al. (2016) and Zawadowicz et al. (2017) (refer to section 1). In addition, alignment of the excimer laser focus in x, y, and z position influences optimum hit rates (Ramisetty et al., 2017)."*

22. Pg8, ln252. Can the authors offer any data to support the conclusion that mass spectral signatures increase with particle size?

*R22: Yes. The signature peaks for 800 nm PSL particles are larger than those from 500 nm particles, as clearly shown in the figures below. They were added to the supporting information as Figure S2.*

**Author comment on "Laser ablation aerosol particle time-of-flight mass spectrometer (LAAPTOF): Performance, reference spectra and classification of atmospheric samples" by Xiaoli Shen et al.**

[Figure]

*Figure S2: Typical spectra with raw signal for individual PSL particles of dva 528 nm (panel A) and 820 nm (panel B), respectively.*

23. Pg8, ln275. Are the authors referring to detector ringing? Detector ringing in MCPs is caused by pulse reflections and therefore should not be counted as ion signal.

*R23: This phenomenon was described by Gross et al. (2000) for high-intensity peaks due to partial saturation of the data acquisition system or signal reflections within the data acquisition circuitry. We added the corresponding literature and rephrased this sentence as following:*

*"Note that the extra peak at m/z 40⁺ besides m/z 39 K⁺ in Fig. 3 (B) is likely due to the incorrect mass assignments as a result of peak shifts (refer to section 2.5 and "Procedure 1" in the supplementary information). For high-intensity peaks such as sodium chloride NaCl, extra peaks next to the main peak (Fig. S3) may have an additional reason: "Ringing" due to partial saturation of the data acquisition system or signal reflections within the data acquisition circuitry (Gross et al., 2000)."*

24. Pg12, ln437. Please explain what 'spectrum-to-spectrum peak shifts' is referring to and how that impacts the assignment of particle type.

*R24: Compare also R14. This is referring to that the peak position for the same ion fragment may shift from spectrum to spectrum, due to the variance in the position of particle-laser interaction.*

*We have revised this sentence as following: "About 10% of the particles cannot be grouped into any type. This is most likely because of an incorrect mass assignment for the stick spectra, resulting from too large spectrum-to-spectrum peak shifts for the same ion fragments/clusters which cannot be corrected on a single particle basis with the existing software (Ramisetty et al., 2017)."*

25. Pg13, ln447. This paragraph contains some nice conclusions about particle classification and should be given more prominence in the manuscript.

*R25: Thank you for your advice. We have given it more prominence in the revised manuscript, especially in the abstract and conclusion part.*

*In the abstract, we have revised the last few sentences as "An exemplary one-day ambient data set was analysed by both classical Fuzzy clustering and a reference spectra based classification method. Resulting identified particle types were generally well correlated. We show how a combination of both methods can greatly improve the interpretation of single particle data in field measurements."*

*In the conclusion, we have revised the penultimate paragraph as "A set of 7314 mass spectra obtained during one day of field measurements was used for particle type classification by both Fuzzy clustering and reference spectra. Fuzzy clustering-yielded six different classes, which could then be identified with the help of reference spectra. Classification of the mass spectra based on comparison with 17 reference spectra resulted in 13 different particle types, six of which exhibited high correlation with the Fuzzy clusters (r: 0.76–0.95). Compared with the reference spectra, we found that each particle class/type has a sulphate signature at m/z 80 SO₃⁻ and 97*

Laser ablation aerosol particle time-of-flight mass spectrometer (LAAPTOF): Performance, reference spectra and classification of atmospheric samples
Xiaoli Shen

**Author comment on "Laser ablation aerosol particle time-of-flight mass spectrometer (LAAPTOF): Performance, reference spectra and classification of atmospheric samples" by Xiaoli Shen et al.**

*$HSO_4^-$, a nitrate signature at m/z 46 $NO_2^-$ and 62 $NO_3^-$, an organic compound signature at m/z 24 $C_2^-$, 25 $C_2H^-$ and 26 $C_2H_2/CN^-$ and a nitrogen-containing organic signature at m/z 26 $CN^-$ and 42 $CNO^-$. Furthermore, we performed a target-oriented classification by using selected reference spectra, allowing for the identification of particles with low number fraction in the ambient aerosol, e.g. lead-containing particles. Based on our results we advise using a combination of both methods for the analysis of SPMS field data."*

26. Pg13, ln454. Only PSL was measured in that size range.

*R26: Yes, you are right. We have revised this sentence as "In this study, the overall detection efficiency (ODE) of LAAPTOF was determined to range from ~(0.01 ± 0.01)% to ~(4.23 ± 2.36)% for polystyrene latex (PSL) with the size of 200 to 2000 nm, ~(0.44 ± 0.19)% to ~(6.57 ± 2.38)% for ammonium nitrate ($NH_4NO_3$) and ~(0.14 ± 0.02)% to ~(1.46 ± 0.08)% for sodium chloride (NaCl) particles in the size rage of 300 to 1000 nm."*

**Author comment on "Laser ablation aerosol particle time-of-flight mass spectrometer (LAAPTOF): Performance, reference spectra and classification of atmospheric samples" by Xiaoli Shen et al.**

Typos/Technical:

27. Pg2, ln 51. Sentence structure.

*R27: We have revised this sentence as "To the best of our knowledge, so far there is no quantitative analysis of particle composition by SPMS, since the ablation/ionization laser cannot interact with the entire particle and the generated ion fragments/clusters are susceptible to matrix effects (Ramisetty et al., 2017)."*

28. Pg 2, ln 55. Home-built.

*R28: Changed as suggested.*

29. Pg3, ln76. 'Number percentage' should be 'number fraction'?

*R29: Changed as suggested.*

30. Pg3, ln104. Remove 'Major'.

*R30: Changed as suggested.*

31. Pg3, ln106. Insert 'and' after SOA.

*R31: Changed as suggested.*

32. Pg3, ln109. Sentence structure.

*R32: We have revised this sentence as "An example for field data analysis based on reference spectra as well as Fuzzy c-means clustering will be given in chapter 3.3."*

33. Pg5, ln194. Check the spacing after the references.

*R33: We have removed extra spacing after the references.*

34. Pg10, ln360. Poor sentence structure.

*R34: We have revised this sentence as "The German soil dust, however, contains more organic species than soil dust from Argentina and USA, reflected in higher intensities at m/z $24^-$, $25^-$, and $26^-$. Argentinian soil dust contains much less mineral species, expressed in much lower intensities of mineral signatures, e.g. m/z $27^+$, $28^+$, $40^+$, $44^+$, and $56^+$."*

**Author comment on "Laser ablation aerosol particle time-of-flight mass spectrometer (LAAPTOF): Performance, reference spectra and classification of atmospheric samples" by Xiaoli Shen et al.**

**Referee #2 comments:** It is my pleasure to provide a review for this manuscript. The paper describes a comprehensive set of test measurements conducted with a new single particle mass spectrometer, the LAAPTOF. The study provides a good overview over the instrument's core capabilities and performance. As there is limited information on the LAAPTOF in existing literature, this study is a much-needed reference for current and future LAAPTOF users, and for the aerosol mass spectrometry community in general. In terms of content, there is a wealth of information in this paper, with little to add or change; except some of the choices in the field data analysis should be explained better. In terms of organization and readability, there is room for improvement: The laboratory measurements presented in the paper were done with several different setups and sample preparation techniques, and, for interpretation, the samples are grouped according to the mixing state of the aerosol. Some of the measurements were size-selective whereas others were not, but for those that weren't, sizes are still reported. Currently, the information which aerosol type was generated how, sampled with what setup, and assigned to which group, is contained in the flow text of section 2.2, and in Table S1 and its footnotes. Both of these are difficult to refer back to quickly, and it is hard for the reader to keep it all straight. The presentation of experimental setups (Figure 1) is also a little confusing, as different laboratory setups are conflated with the field setup (making it look like one experimental setup even though it is not) and simple schematics are combined with a detailed functional sketch of the LAAPTOF instrument. I recommend final publication after consideration of the following suggestions and comments:

1. The correlation method (a consistent name should be used for this method throughout the paper) should be explained better. Why were positive and negative spectra treated separately (10 and 7, respectively)? Aren't the positive-negative pairs intimately connected, coming from one particle? How were they re-combined to yield 13 particle types (Figure 10)? The caption of Figure S13 contains hints, but is extremely hard to understand. Also, referring to lines 371 – 372 (and the procedure description in the supplement): there are ways to determine the appropriate number of clusters aside from "experience of the operating scientist" – see for example Hinz et al. (1999). Is there a specific reason these were not used?

*R 1: 1) We revised the text and use the name "Reference spectra based classification" throughout the main manuscript as well as the supplement.*

*2) The positive and negative spectra were treated separately due to differences in peak intensities for the two polarities, which biases calculated coefficients r; namely high intensive peaks dominating the value of r. However, this can be avoided by selecting specific mass ranges for the correlation. We have added this explanation in the supplement under Procedure 2.*

*3) . The individual steps of the the reference spectra based classification method are explained in the supplement in paragraph Procedure 2, which now reads as following:*

*"Step 1: Normalize (to sum) stick spectra from the field data and reference spectra.*

*Step 2: Correlate each ambient spectrum with each reference spectrum (positive spectra 01 to 32 and negative spectra 01 to 32). Positive and negative spectra are treated separately due to differences in peak intensities for the two polarities, biasing coefficients r (refer to equation S1); namely high intensive peaks dominating the value of r. However, this can be reduced by selecting specific mass ranges for the correlation.*

*Step 3: Define particle type. Only correlations with a Pearson's correlation coefficient $r \geqslant 0.6$ are used. Particles are categorized using a 4-digit code: The first 2 digits identify the positive reference spectrum (01 – 32) with which a particle's spectrum exhibits maximum r; the third and fourth digits the negative reference spectrum (01 – 32) with which the particle's negative spectrum exhibits maximum r.*

*Step 4: Reduce the number of references. A histogram with the number of particles per particle type is used to identify the main particle types. For positive spectra, particle types with number fractions $\geqslant 1\%$ were chosen. For the negative spectra, no numerical criteria could be used in the first iteration, but spectra were examined individually (we found e.g. that four arable soil dust samples had similar negative spectra and only used SDGe01). For our dataset the number of reference spectra or particle types was reduced to 10 positive and 7 negative spectra (Table S2).*

*Step 5: Repeat step 1 to 3 using the reduced set of reference spectra.*

*Step 6: Identify the main particle types. Categorized particles are again plotted in a histogram, and main particle types are identified based on number fractions ≥ 1%. Only particles with correlations in both polarities are included here. In our study, 13 main types were found."*

*4) Regarding the Fuzzy clustering procedure, you are right, there are some criteria to determine the optimal number of clusters. We chose the "the most straightforward and intuitive approach" as described by Hinz et al. (1999), which "is to compare the graphical plots of the cluster centres for the different numbers of clusters. This procedure requires the manual recognition of higher and lower degrees of grouping and separation of elements (mass spectra)".*

2. How was the particle size information obtained and handled? According Section 3.2, line 246, a $d_m$ of 800 nm was chosen (presumably only for the samples passing through the DMA). In Table S1, this size only appears for PSL, and other sizes ($d_{va}$) are given for all samples. One of the footnotes mentions a Gaussian fitting, but this should be described more clearly in the "methods" section.

*R 2: For the particles generated from a nebulizer and passed through a DMA for sizing, we chose a $d_m$ of 800 nm. This size was chosen for all the samples (not only for PSL) generated by this method. We have added this information to the footnotes of Table S1. For some polydisperse particle samples size information ($d_m$) was obtained by SMPS measurements, e.g. SOA generated in the AIDA chamber. For e.g. particles mobilized by shaking their reservoir bottles, no SMPS measurements were done. We obtained $d_{va}$ by the LAAPTOF measurements for all samples (given in Table S1).*

*We have added a more clear description of particle sizing by LAAPTOF, including the Gaussian fitting, in section 2.5 Spectral and size data analysis: "In addition, particle size ($d_{va}$) was recorded for individual particles. The corresponding size distribution can be plotted as $d_{va}$ histogram, a Gaussian fit of which yields number mean $d_{va}$ values and the standard deviation (width)."*

3. Organization: There are three experiments here: 1. a rigorous determination of ODE for some aerosol types, 2. a comprehensive collection of reference spectra, 3. a field study with two different data analysis methods. For the sake of readability, the three should be treated separately as much as possible. Here are some suggestions to improve on clarity:

a) Move table S1 into the main manuscript. In the footnotes, make sure that the explanations for the generation methods are easy to spot – perhaps a list, rather than wrapped into a sentence.

*R 3 a): We have accounted for this by moving Table S1 to the main manuscript as Table 1 and listed the generation methods in the footnotes.*

b) The three (or four, including sampling from reservoir headspaces?) experimental setups should be defined in Section 2 (separately from the generation methods) and referred back to throughout the paper. Generation methods A, B1, B2, and S should also be defined in the text in Section 2, and referred back to.

*R 3 b): We have separately defined the generation methods and set ups in Section 2.2 as suggested. We made sure all the corresponding contents can be referred back to.*

*The revised contents are as follows:*

*"**2.2 Aerosol particle generation and experimental set up in the laboratory***

*The laboratory based aerosol particles measured in this study (summarized in Table 1) were generated in four different ways (A, B1, B2, and S).*

*Method A: Samples for pure particles and homogeneous and heterogeneous mixtures were dissolved/suspended in purified water and nebulized (ATM 221; Topas GmbH) with dry synthetic air, passed through two diffusion dryers (cylinder filled with Silica gel, Topas GmbH), and then size selected by a Differential Mobility Analyser (DMA 3080, TSI GmbH) before being sampled by LAAPTOF.*

*Method B1: Particles were sampled from the 84.5 $m^3$ simulation chamber AIDA (Aerosol Interactions and*

*Dynamics in the Atmosphere) of KIT (Saathoff et al., 2003). SOA particles were formed in the 3.7 m³ stainless steel Aerosol Preparation and Characterization (APC) chamber via ozonolysis (~6 ppm ozone) of α-pinene (~2.2 ppm) and then transferred into AIDA. Soil dust samples were dispersed by a rotating brush generator (RGB1000, PALAS) and injected via cyclones into the AIDA chamber. Sea salt particles were generated and injected into AIDA by ultrasonically nebulizingartificial seawater (Sigma Aldrich) and highly concentrated skeletonema marinoi culture (in artificial seawater), respectively, via a droplet separator and 2 diffusion dryers (Wagner et al., 2017).*

*Method B2: Used only for soot particles, which were generated with a propane burner (RSG miniCAST; Jing Ltd.) and injected into and sampled from a stainless steel cylinder of 0.2 m³ volume.*

*Method S: Silica, Hematite, Illite_NX, Arizona test dust, desert and urban dust, black carbon from Chestnut wood (University of Zürich, Switzerland), and diesel soot reference particles from NIST were suspended in their reservoir bottles by shaking them and sampled directly from the headspace (upper part) of these reservoirs through a tube connecting it with the LAAPTOF.*

*For all the measurements, except measuring the method S-generated particles, a condensation particle counter (CPC 3010, TSI GmbH) was used to record the particle number concentration in parallel with the LAAPTOF inlet. Setup in Fig. 1 was specific for particles generated from method A."*

c) My suggestion for Figure 1 would be to only show setup A, as there is no information in setups B and C that cannot be described easily in the text. Setup A should then be shown in more detail (flow rates, flow splitting between LAAPTOF inlet and CPC, etc.). This would make it easier for the reader to mentally separate the ODE experiments from the reference spectra experiments. Also, in the Figure, a box or some other designation should be put around the actual LAAPTOF instrument. One could also show setup B in a separate figure, with more detail on the particle generation (cyclones, APC chamber, CAST, etc.), but I believe it is not necessary, as there are descriptions and references in section 2.2.

*R 3 c): We modified Figure 1 as suggested. The description of methods B and S is given in the text of section 2.2, and we moved the description for setup C to section 2.3 Field measurement, such that lab and field measurements are now separated.*

d) Section 3.1. should be reorganized. Lines 167 – 174 in Section 3.1.1 should be part of Section 2 (Methods). Section 3.1.2 is currently a mix of discussion of ODE measurement results, interpretation of features in the reference spectra, and literature review. While the factors mentioned in lines 223 - 239 may influence ODE, their impact is not put into quantitative relation to the presented measurement results. The discussion would be more appropriately placed in the introduction (some of it, such as the40 mW laser setting, in Section 2.1). I would also argue that lines 211 – 221 could be discussed in the context of the reference spectra (3.2); at least, though, there should be several references to Section 3.2.

*R 3 d): We agree and have moved lines 167-174 in Section 3.1.1 to the Methods section, as section 2.4 Efficiency calculations.*

*Regarding lines 223-239 in Section 3.1.2, we deleted the redundant content They now read as follows: "The incident intensity of radiation, which is another parameter that influences the light scattered by particles (as well as background signal caused by stray light), is related to power and beam dimensions of the detection laser. Corresponding instrument modifications were done by Marsden et al. (2016) and Zawadowicz et al. (2017) (refer to section 1). In addition, alignment of the excimer laser focus in x, y, and z position influences optimum hit rates (Ramisetty et al., 2017)." Although the influence of some factors such as surface roughness of the particles is not put into quantitative relation to our results, we still leave them in this section. This provides readers with a relatively complete picture of the factors that influence the LAAPTOF performance.*

*Regarding lines 211 -221 in Section 3.1.2, we have added more references as suggested. They are rephrased as:*

*"Optical properties of the particles have a strong impact on how light is scattered and absorbed, and thus it*

*should be noted that the optical properties do not only influence scattering efficiency, but also absorption and ionization efficiency (or hit rate). As shown in Fig. 2F, ODE for $NH_4NO_3$ is higher than that for NaCl at any size we studied. This is mainly caused by differences in their optical properties of scattering. Relative fresh soot particles scatter only little light due to their black colour and small size (typically ~ 20 nm) of the primary particles forming their agglomerates, and are thus hardly detected by the detection laser. However they are good light absorbers and thus relatively easy to ablate and ionize. The reference spectra of pure $NH_4NO_3$ and $(NH_4)_2SO_4$ particles showed intensive prominent peaks for pure $NH_4NO_3$ particles (refer to Fig. 3A) but only one weak peak of m/z 30 $NO^+$ for pure $(NH_4)_2SO_4$ particles. This indicates that $NH_4NO_3$ is a better absorber than $(NH_4)_2SO_4$, and thus easier to ablate and ionize. For homogeneous mixtures of these two ammonium salts, the sulphate species are ablated and ionized much more easily (refer to section 3.2.2), due to increased UV light absorption by the nitrate component. Some small organic compounds with weak absorption properties are hard to ablate and ionize, e.g. oxalic acid ($C_2H_2O_4$), pinic acid, and cis-pinonic acid. They exhibited much weaker signals (~80% lower) than macromolecular organic compounds in PSL or humic acid particles."*

f) The data analysis methods should be presented in a separate subsection of Section 2. The sections headed "Procedures for ..." and "Seeking..." in the supplement (which could be very useful for other LAAPTOF users), should be proofread (informal expressions), designated ("Procedure 1", "Procedure 2", or something similar) and referred to in Section 2.

*R 3 f): We have put the data analysis methods in a separated subsection 2.5. In the supplement, we revised the corresponding titles as "Procedure 1: LAAPTOF data analysis with emphasize on mass calibration", "Procedure 2: Reference spectra based classification" and "Procedure 3: Seeking lead (Pb) containing particles using reference spectra". "Procedure 1" "Procedure 2" and "Procedure 3" are used in the main manuscript to refer to the supplement.*

*The new subsection we added reads as follows:*

*"**2.5 Spectral and size data analysis***

*For each type of laboratory generated aerosol particle, we measured at least 300 mass spectra. Data analysis is done via the LAAPTOF Data Analysis Igor software (Version 1.0.2, Aeromegt GmbH). There are five main steps for the basic analysis procedure: a) removal of the excimer laser ringing signal from the raw mass spectra; b) determination of the signal baseline; c) filtering for empty spectra; d) mass calibration; and e) stick integration. Spectrum-to-spectrum differences in peak positions for the same ion fragments/clusters, due to variance in the position of particle-laser interaction, complicate the mass calibrations. This can't be corrected with the existing Aeromegt software (Ramisetty et al., 2017). Details can be found in "Procedure 1" in the supplementary information. Spectra presented in this paper were typically normalized to the sum of ion signal before further aggregation.*

*For ambient data analysis, we used two different classification methods. The first one is Fuzzy c–means clustering algorithm embedded in the LAAPTOF Data Analysis Igor software, commonly used to do classification based on the similarities of the individual spectra. The number of the classes is chosen manually, afterwards the particle spectra with a minimum distance between their data vectors and a cluster centre will be grouped into a specific class (Hinz et al., 1999; Reitz et al., 2016). Since each spectrum can belong to multiple classes (Reitz et al., 2016) the resulting fraction/percentage for each class represents the information about the degree of similarity between aerosol particles in one particular class, and not a number percentage. The second method developed in this study is based on the correlation between each ambient spectrum and our reference spectra. The resulting Pearson's correlation coefficient (r) is used as the criterion to group particles into different types (here we use "types" instead of "classes" in order to differentiate these two classification methods). When r is higher than a threshold value of 0.6, the ambient spectrum is considered to have high correlation with the corresponding reference spectrum. For simplification we chose 10 positive and 7 negative reference spectra. For example, we only use German soil dust as the reference for arable soil dust rather than using four arable soil dust samples from different places. More details about the procedure for this method as*

**Author comment on "Laser ablation aerosol particle time-of-flight mass spectrometer (LAAPTOF): Performance, reference spectra and classification of atmospheric samples" by Xiaoli Shen et al.**

*well as the corresponding equations and uncertainties estimation can be found in "Procedure 2" in the supplementary information.*

*In addition, particle size ($d_{va}$) was recorded for individual particles. The corresponding size distribution can be plotted as $d_{va}$ histogram, a Gaussian fit of which yields number mean $d_{va}$ values and the standard deviation (width)."*

4. Presentation of Figures: Figure 2: Panel A is very important, but hard to read. Consider putting the y-axis on a linear scale and repeating the x-axis on the top of the graph. Figures 4-7: It is hard for the reader to go back between label shadings in the various panels and description of the shadings in the caption. A legend for the shading in or next to each panel would be much more convenient. What are the dva values in Figure 4 referring to exactly? Figures 4 C (+), and 7: It would be nice to have a few more peak labels (ions, not just m/z values). Figure 9: As much as possible, avoid extra abbreviations, they are not reader-friendly. For example, there is no reason that the axis tick label "SC" could not be "NaCl". Figure 10: In the current layout, this Figure is too small.

*R 4: We have changed these figures as suggested.*

**Other comments:**

5. lines 82-86: There is another recent paper describing LAAPTOF spectra by Wonaschuetz et al. (2017, Journal of Aerosol Science 113, 242 - 249)

*R 5: We refer to this new paper now in the introduction and method section of the manuscript.*

6. line 124: Can provide a short description of what exactly is integrated in "stick integration"?

*R 6: We have added a short description of "stick integration", that is the integration of nominal masses for peaks.*

7. line 130-131, explanation of percentages: Is this quantity ever used in the paper? If not, this explanation could be omitted.

*R 7: We use the percentages in section 3.3 Interpretation of field data.*

8. line 141: Consider "dissolved/suspended" (PSL spheres don't dissolve)

*R 8: Changed as suggested.*

9. line 149: Can you add a very short description of the "different ways"?

*R 9: We have added a short description in the manuscript, as follows: "Pure and organics containing sea salt particles were generated and injected into AIDA by ultrasonically nebulizingartificial seawater (Sigma Aldrich) and highly concentrated skeletonema marinoi culture (in artificial seawater), respectively, via a droplet separator and 2 diffusion dryers (Wagner et al., 2017)."*

10. line 161: How long was the LAAPTOF deployed?

*R 10: LAAPTOF was deployed for ~5 weeks from July 26 to August 31. We have added this information in the manuscript.*

11. line 173: In the samples generated by nebulizing salts, followed by size selection in the DMA, how was the possibility of multiply charged particles handled?

*R 11: Of course this method can potentially produce e.g. doubly charged particles with larger geometrical size. However, checking the $d_{va}$ size distributions for them shows no significant contribution.*

12. line 200: Can you provide a reference?

*R 12: Yes, we have added one reference (Schreiner et al., 1999).*

13. line 203: It seems that it would be easy to provide a quick comparison of Mie scattering efficiencies at the particle sizes/laser wavelength relevant to this measurement to validate the M-shape.

**Author comment on "Laser ablation aerosol particle time-of-flight mass spectrometer (LAAPTOF): Performance, reference spectra and classification of atmospheric samples" by Xiaoli Shen et al.**

*R 13:Yes, you are right. We have added a new figure in the supporting information to show the quick comparison as suggested. The revised sentences in the main manuscript are as follows:*

*"The scattering efficiencies of PSL particles, based on Mie calculation at the particle sizes and detection laser wavelength relevant to our LAAPTOF measurement, validate the M-like shape of $SE_{PSL}$ (refer to Fig. S1)."*

[Figure]

*Figure S1: Comparison of scattering efficiencies for PSL particles measured by LAAPTOF and total scattering efficiencies from Mie calculations as a function of particle size. The input parameters for Mie calculation are a refractive index of 1.62 and a wavelength of 405 nm.*

14. line 234: Was this controlled for in some way in the test measurements?

*R 14: Yes, we controlled the number concentrations by controlling the amount of the original samples dissolved/suspended in purified water, which were nebulized with dry synthetic air for LAAPTOF measurements.*

*Gemayel et al. (2016) reported the upper limits of the incoming particles, which ranged from 620 to 450 particles $cm^{-3}$ for 350 to 700 nm PSL particle.*

15. line 271: This is confusing. Which ratio is 1?

*R 15: The ratio of m/z 76 $SiO_3^-$ + m/z 77 $HSiO_3^-$ to m/z 60 $SiO_2^-$ is 1. We have revised the sentence as: "Another inorganic compound measured here is silica (Fig. S4) and its typical peak ratio of (m/z 76 $SiO_3^-$ + m/z 77 $HSiO_3^-$) to m/z 60 $SiO_2^-$ is ~1.0"*

16. line 285: Can you provide a reference?

*R 16: We have revised this sentence as :" In addition, as already discussed in section 3.1.2, the better UV light absorber $NH_4NO_3$ assists in light absorbing for mixed particles, resulting in a sulphate signature that could not be observed for pure $(NH_4)_2SO_4$."*

Minor corrections/typos:

17. line 296: "can be found".
*R 17: Changed as suggested.*
18. line 305: "are weaker"
*R 18: Changed as suggested.*
19. line 328: "panel A"
*R 19: Changed as suggested.*
20. line 312: "are observed"
*R 20: Changed as suggested.*
21. line 332 – 337: This is a very long sentence. Consider this construction: "In spectra (A), we also observed the following peaks: ...."
*R 21: Changed as suggested.*
22. line 339: consider: "Most of the ... fragments of soil dust are similar to those of desert dust..."
*R 22: Changed as suggested.*
23. line 348: Start the sentence with "The" and remove the "s" in "soil dust(s)" from Argentina".

**Author comment on "Laser ablation aerosol particle time-of-flight mass spectrometer (LAAPTOF): Performance, reference spectra and classification of atmospheric samples" by Xiaoli Shen et al.**

*R 23: Changed as suggested.*

24. line 454 – 456: "e.g." is in a strange place in the sentence. Perhaps: "...from different fragments: for example, m/z 26 ...."

*R 24: Changed as suggested.*

25. line 442: "In any case" seems to be a needless expression.

*R 25: We have removed "In any case".*

26. line 467: consider replacing "among the huge amount of ambient data" by "in the ambient aerosol"

*R 26: Changed as suggested.*

*References:*

*Gemayel, R., Hellebust, S., Temime-Roussel, B., Hayeck, N., Van Elteren, J. T., Wortham, H., and Gligorovski, S.: The performance and the characterization of laser ablation aerosol particle time-of-flight mass spectrometry (LAAP-ToF-MS), Atmos Meas Tech, 9, 1947–1959, 2016.*

*Gross, D. S., Gälli, M. E., Silva, P. J., and Prather, K. A.: Relative sensitivity factors for alkali metal and ammonium cations in single particle aerosol time-of-flight mass spectra, Anal Chem, 72, 416–422, 2000.*

*Hinz, K. P., Greweling, M., Drews, F., and Spengler, B.: Data processing in on-line laser mass spectrometry of inorganic, organic, or biological airborne particles, J Am Soc Mass Spectr, 10, 648–660, 1999.*

*Kulkarni, P., Baron, P. A., and Willeke, K.: Aerosol measurement: Principles,techniques, and applications, John Wiley & Sons, Inc., 2011.*

*Li, L., Huang, Z. X., Dong, J. G., Li, M., Gao, W., Nian, H. Q., Fu, Z., Zhang, G. H., Bi, X. H., Cheng, P., and Zhou, Z.: Real time bipolar time-of-flight mass spectrometer for analyzing single aerosol particles, Int J Mass Spectrom, 303, 118–124, 2011.*

*Lin, Q. H., Zhang, G. H., Peng, L., Bi, X. H., Wang, X. M., Brechtel, F. J., Li, M., Chen, D. H., Peng, P. A., Sheng, G. Y., and Zhou, Z.: In situ chemical composition measurement of individual cloud residue particles at a mountain site, southern China, Atmos Chem Phys, 17, 2017.*

*Marsden, N., Flynn, M. J., Taylor, J. W., Allan, J. D., and Coe, H.: Evaluating the influence of laser wavelength and detection stage geometry on optical detection efficiency in a single-particle mass spectrometer, Atmos Meas Tech, 9, 6051–6068, 2016.*

*Pöschl, U.: Atmospheric aerosols: Composition, transformation, climate and health effects, Angew Chem Int Edit, 44, 7520–7540, 2005.*

*Ramisetty, R., Abdelmonem, A., Shen, X., Saathoff, H., Mohr, C., and Leisner, T.: Comparison of nanosecond and femtosecond laser ablation in single particle mass spectrometer for atmospheric applications, Atmos Meas Tech Discuss, doi.org/10.5194/amt-2017-357, 2017.*

*Schreiner, J., Schild, U., Voigt, C., and Mauersberger, K.: Focusing of aerosols into a particle beam at pressures from 10 to 150 Torr, Aerosol Sci Tech, 31, 373–382, 1999.*

*Zawadowicz, M. A., Abdelmonem, A., Mohr, C., Saathoff, H., Froyd, K. D., Murphy, D. M., LeisnerI, T., and Cziczo, D. J.: Single-particle time-of-flight mass spectrometry utilizing a femtosecond desorption and ionization laser, Anal Chem, 87, 12221–12229, 2015.*

*Zawadowicz, M. A., Lance, S., Jayne, J. T., Croteau, P., Worsnop, D. R., Mahrt, F., Leisner, T., and Cziczo, D. J.: Quantifying and improving the performance of the Laser Ablation Aerosol Particle Time of Flight Mass Spectrometer (LAAPToF) instrument, Atmos Meas Tech Discuss, doi: 10.5194/amt-2017-1, 2017.*

*Best wishes,*

*Xiaoli Shen and co-authors*